# Local Distribution-Based Adaptive Oversampling for Imbalanced Regression

**Shayan Alahyari**                                                    *salahya@uwo.ca*
*Western University*

**Mike Domaratzki**                                                    *mdomarat@uwo.ca*
*Western University*

**Reviewed on OpenReview:** *https://openreview.net/forum?id=6qYTR9iJdm*

## Abstract

Imbalanced regression occurs when continuous target variables have skewed distributions, creating sparse regions that are difficult for machine learning models to predict accurately. This issue particularly affects neural networks, which often struggle with imbalanced data. While class imbalance in classification has been extensively studied, imbalanced regression remains relatively unexplored, with few effective solutions. Existing approaches often rely on arbitrary thresholds to categorize samples as rare or frequent, overlooking the continuous spectrum of target values. These methods can produce synthetic samples that fail to improve model performance and may discard valuable information through undersampling. To address these limitations, we propose LDAO (Local Distribution-based Adaptive Oversampling), a novel data-level approach that avoids categorizing individual samples as rare or frequent. Instead, LDAO learns the global distribution structure by decomposing the dataset into a mixture of local distributions, each preserving its statistical characteristics. LDAO then models and samples from each local distribution independently before merging them into a balanced training set. LDAO achieves a balanced representation across the entire target range while preserving the inherent statistical structure within each local distribution. In extensive evaluations on 45 imbalanced datasets, LDAO outperforms state-of-the-art oversampling methods on both frequent and rare target values, demonstrating its effectiveness for addressing the challenge of imbalanced regression. Our code is available at https://github.com/ShayanAlahyari/LDAO.

## 1 Introduction

In classification tasks, imbalance occurs when some classes have far fewer samples than others, in both binary and multi-class settings (He & Garcia, 2009; Guo et al., 2017). The minority classes are overshadowed by the majority classes, causing traditional classifiers to focus on the more abundant classes and perform poorly in identifying minority classes (Chawla et al., 2002). This leads to inaccurate detection of minority classes since models struggle to recognize patterns in sparsely sampled regions (Buda et al., 2018). Critical applications like fraud detection, medical diagnostics, and fault detection are particularly affected, as rare but important classes might be missed (Johnson & Khoshgoftaar, 2019). Therefore, effective handling of imbalanced data is crucial for reliable performance across all classes, ensuring significant rare examples are properly detected (Liu et al., 2009).

While imbalance is widely studied in classification (He & Garcia, 2009), it also affects regression tasks that predict continuous values (Krawczyk, 2016). Imbalanced regression occurs when certain target ranges (typically rare or extreme cases) have significantly fewer samples than others (Branco et al., 2016; Torgo et al., 2013). Traditional regression models struggle to accurately predict these rare values because they focus on more frequent, well-represented ranges (Chawla et al., 2004; Branco et al., 2016). Accurately predicting

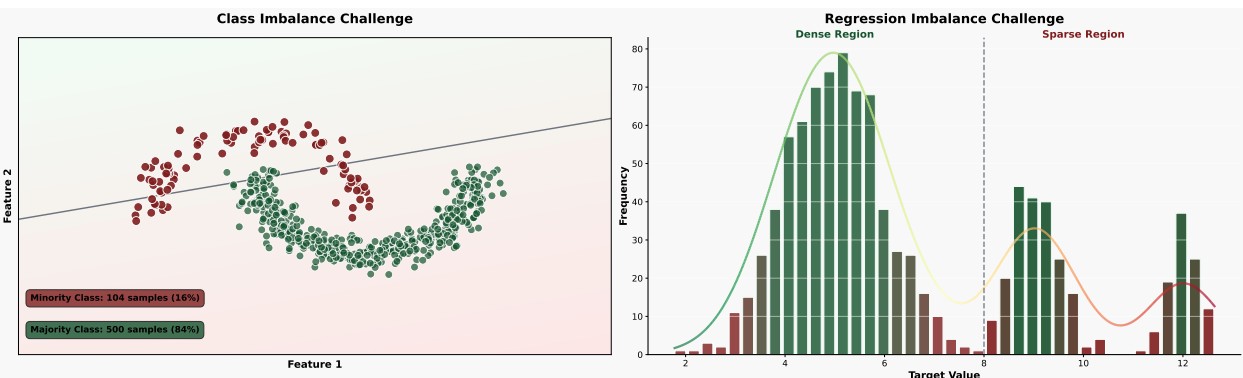

Figure 1: Left: a long-tailed classification problem in which discrete class counts allow straightforward identification of minority (red) and majority (green) classes. Right: an imbalanced regression problem in which continuous target values create underrepresented regions (red) that cannot be detected by simple counting. Both settings can exhibit long-tailed distributions; here we emphasize their difference in minority detection mechanisms.

these less frequent target ranges is challenging yet crucial for many real-world applications (Torgo et al., 2013).

Figure 1 contrasts imbalanced classification with imbalanced regression. In classification, identifying minority classes is straightforward because labels are discrete and can be counted. In regression, minority regions exist within continuous target distributions, often in sparsely populated tails. While classification imbalance relates to distinct labels, regression imbalance depends on distribution shape (skewed or multi-modal). Classification methods cannot be directly applied to regression since there is no simple way to count labels in order to identify underrepresented values. This makes detecting and addressing sparse regions a significant challenge in imbalanced regression.

Imbalanced regression impacts many critical real-world applications where accurately predicting rare but high-impact events in the tails of skewed distributions is paramount. In electrochemical prognostics, lithium-ion battery state-of-health data exhibit long-tailed degradation trajectories, with very few cells nearing end-of-life available for training, leading to significant underestimation of remaining useful life by standard models (Zhao et al., 2025). In automated pain assessment, continuous video-based pain intensity labels are overwhelmingly concentrated in low-pain frames, causing deep facial expression regression networks to underpredict severe pain episodes (Xiang et al., 2022). In carbon markets, sporadic extreme price surges in emissions trading schemes create highly imbalanced time-series segments that traditional regressors fail to capture, resulting in substantial forecasting errors during critical market movements (Yin & et al., 2025). In fuzzy-rule-based optimization, conventional soft computing methods disregard the uneven distribution of target values, undermining regression performance on rare but crucial value ranges (Arteaga-Jover et al., 2023). These examples represent only a subset of numerous instances in which addressing imbalance in regression is essential.

Several solutions have been proposed to address imbalanced regression at both data and algorithmic levels. Some extend SMOTE from classification to regression by generating synthetic examples in the target distribution's tails, increasing representation of rare values (Torgo et al., 2013; Branco et al., 2017). Others apply cost-sensitive methods that penalize errors on underrepresented values more heavily, encouraging models to focus on these instances (Steininger et al., 2021). Many data-level approaches rely on linear interpolation which is adopted from SMOTE, where a synthetic sample is generated between a minority sample $\mathbf{x}_i$ and a neighbor $\mathbf{x}_j$:

$$\mathbf{x}_{\text{new}} = \mathbf{x}_i + \lambda\,(\mathbf{x}_j - \mathbf{x}_i), \quad \lambda \sim \mathcal{U}(0, 1).$$

This is problematic because it assumes linear relationships between data points, which often fail to capture the complex, non-linear structure of the true data distribution. The addition of basic Gaussian noise is

another common strategy, but this method is often insufficient for creating realistic samples that follow the intricate patterns of the true feature-target relationship.

These synthetic samples often fail to accurately represent the true, complex relationship between the input features and the continuous target variable, which distorts the joint feature-target distribution. Furthermore, the complex web of correlations between different features is frequently lost entirely because the generation process is often univariate or does not explicitly model these relationships. The generated data can also fail to reproduce the basic statistical profile of the real data, exhibiting incorrect statistical properties where the marginal distributions of the synthetic samples have incorrect moments, such as mean and variance, or shapes, such as skewness and kurtosis, compared to the original data.

We propose LDAO, a novel data-level oversampling approach designed to directly overcome these fundamental limitations through adaptive, distribution-aware sampling. Instead of imposing simplistic linear assumptions, LDAO operates on the joint feature-target space, first decomposing the data into a mixture of local distributions to capture its global structure. Within each discovered region, LDAO employs multivariate Kernel Density Estimation (KDE) to build a rich, non-parametric model that explicitly learns the local covariance structure. By generating new samples from these localized models, LDAO ensures that synthetic data preserves the intricate feature-target dependencies, inter-feature correlations, and the essential statistical properties of the original data. This method maintains the continuity of regression targets while providing a statistically sound and robust foundation for training regression models on imbalanced data.

## 2 Related Work

Adapting data-level oversampling strategies to regression is more difficult because the target variable is continuous (He & Ma, 2013; Krawczyk, 2016). Early approaches to imbalanced regression modified classification resampling by introducing a relevance function $\phi(y)$ to distinguish between rare and frequent target regions (Torgo & Ribeiro, 2007). This function assigns higher scores to more extreme target values, while a user-specified threshold determines which points are rare versus common (Torgo & Ribeiro, 2007; Ribeiro, 2011). Using this relevance function framework, Torgo et al. developed two main resampling methods for regression (Torgo et al., 2013; 2015).

The first method applies random undersampling to remove samples with low $\phi(y)$ scores (common target values), reducing their overrepresentation. The second method, SMOTER, generates new samples in regions with high $\phi(y)$ scores (rare target values) by interpolating feature values of nearby rare samples and calculating target values as weighted averages of these neighbors (Torgo et al., 2013; 2015). While these approaches were pioneering in addressing regression imbalance and demonstrated that focusing on rare targets improves performance (Torgo et al., 2015), they depend on user-chosen rarity thresholds that may be arbitrary and fail to reflect the true structure of continuous target distributions (Branco et al., 2019).

SMOGN (Synthetic Minority Over-sampling Technique for Regression with Gaussian Noise) extends SMOTER by combining interpolation with noise-based oversampling. For each rare instance, SMOGN generates either an interpolated synthetic point (as in SMOTER) or adds Gaussian noise to create a perturbed point (Branco et al., 2017). This hybrid approach provides more flexible oversampling of sparse regions and outperforms SMOTER (Branco et al., 2017). SMOGN also applies undersampling to reduce overrepresented frequent examples, establishing it as a state-of-the-art resampling method for imbalanced regression (Branco et al., 2017).

More recently, Camacho et al. (2022) developed Geometric SMOTE (G-SMOTE) for regression, adapting geometrically enhanced SMOTE to continuous targets. G-SMOTE generates synthetic samples in minority regions by considering both feature-space interpolation and target distribution geometry, offering greater flexibility in sample placement (Camacho et al., 2022).

Stocksieker et al. (2023) combined weighted resampling and data augmentation to better represent covariate distributions and reduce overfitting. Their approach weights underrepresented regions while using noise addition or interpolation to cover wider data ranges. Camacho and Bacao introduced WSMOTER (Weighting SMOTE for Regression), replacing fixed thresholds with instance-based weighting to better highlight underrepresented targets and improve synthetic sample generation in sparse regions (Camacho & Bacao,

2024). Stocksieker et al. (2024) later developed GOLIATH, a framework handling both noise and interpolation methods with hybrid generators and wild-bootstrap steps for continuous targets. Aleksic and García-Remesal proposed SUS (Selective Under-Sampling), which selectively removes only those majority samples that provide little additional information, along with an iterative variant (SUSiter) that reintroduces discarded samples over time, showing strong results on high-dimensional datasets (Aleksic & García-Remesal, 2025).

In addition to data-level strategies, algorithm-level solutions embed imbalance handling directly into the learning process. Cost-sensitive learning exemplifies this approach by modifying loss functions or instance weights to make models focus on rare targets and incur higher penalties for errors on these examples. This concept originated in classification by weighting classes inversely to their frequency and extends naturally to regression by emphasizing rare or extreme target values (Zhou & Liu, 2010; Elkan, 2001; Domingos, 1999). By imposing higher costs on underrepresented outcomes, these approaches drive models to reduce errors where accuracy matters most.

DenseLoss is a cost-sensitive approach designed specifically for imbalanced regression (Steininger et al., 2021). It uses DenseWeight, a density-based weighting strategy that assigns higher weights to samples with less frequent target values. This directs the model to focus on rare or extreme cases where accuracy is most critical (Steininger et al., 2021). DenseWeight estimates the target distribution's probability density and gives larger weights to low-density targets, scaling the loss function during training to emphasize prediction accuracy on rare cases. Unlike oversampling methods, DenseLoss modifies the learning process without creating or removing examples from the training set.

Yang et al. (2021) proposed a framework for deep imbalanced regression that improves prediction performance for extreme target values using label-distribution smoothing and adaptive sampling. Their work demonstrates how continuous target imbalance can severely impact deep neural network performance. Ren et al. (2022) introduced Balanced MSE, which modifies standard mean-squared error to give higher weight to errors on rare targets. Validated across multiple computer vision tasks, their approach shows that directly addressing label imbalance outperforms standard MSE in skewed settings.

Deep imbalanced regression (DIR), particularly in vision applications such as age estimation and depth prediction, has recently seen a surge of methods that integrate representation learning and classification paradigms. Dong et al. (2025) propose geometric constraints, enveloping and homogeneity losses, within a surrogate-driven representation learning (SRL) framework to enforce uniform coverage and smooth transitions in the latent space, yielding significant improvements on several DIR benchmarks. Xiong & Yao (2024) reformulate regression as hierarchical classification, introducing hierarchical classification adjustment (HCA) and range-preserving distillation (HCA-d) to balance quantization error and classifier accuracy across coarse-to-fine label partitions. Nejjar et al. (2024) shift from in-weight to in-context learning, demonstrating both theoretically and empirically that selecting localized prompts mitigates majority-region bias and enhances transformer-based DIR performance. Pu et al. (2025) adopt a divide-and-conquer strategy by aggregating nearby labels into ordinal groups, leveraging group-aware contrastive learning, symmetric descending soft labeling, and a multi-expert regressor to preserve label continuity while addressing imbalance in skewed distributions.

## 3 Motivation

Despite recent advances, imbalanced regression still faces significant challenges. Current methods often identify samples as rare or frequent and oversample the distribution based on that assumption. In many cases, these assumptions might be oversimplifications due to the complexity of continuous distributions and might not help the model generalize better or improve its prediction on rare samples. Also, many approaches try to oversample rare targets and undersample frequent targets, which misrepresents the dataset's underlying characteristics. Removing information from datasets and labeling certain ranges as rare or frequent is problematic, as rare samples can occur in any range within the global distribution, especially in multi-modal distributions. Often, distributions have latent characteristics which make it difficult to simply classify which samples are rare or frequent.

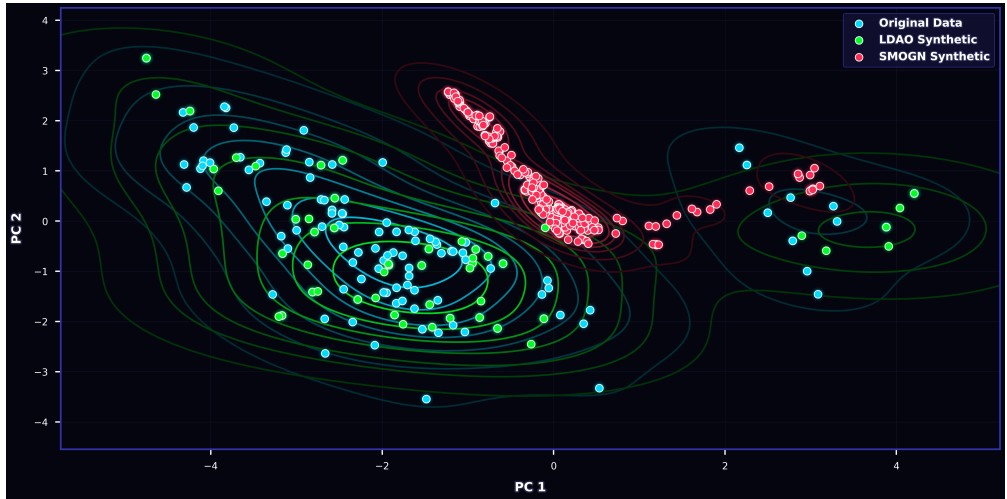

Figure 2: PCA projection of original (cyan), SMOGN synthetic (red), and LDAO synthetic (green) samples for the $\geq$ 75th-percentile tail, showing SMOGN's narrow interpolation ridge versus LDAO's natural dispersion around the extreme values for the Boston dataset (Harrison & Rubinfeld, 1978).

In addition, interpolation and noise-based oversampling techniques may generate synthetic samples that don't reflect actual data patterns, especially in extremely sparse regions (Branco et al., 2019). Figure 2 illustrates this effect: SMOGN synthetic samples (red) lie along a narrow interpolation ridge, while LDAO samples (green) exhibit a broader, more natural dispersion around the extreme tail. Creating synthetic feature-target pairs that remain consistent with the original distribution is challenging, and most methods focus narrowly on extreme tails rather than the entire distribution. Many approaches rely heavily on relevance functions and thresholds that require careful parameter selection. Poor choices can lead to oversampling marginally rare areas while missing truly rare regions. This dependence on domain-specific tuning limits broader applicability across different datasets. Real-world applications in healthcare, genomics, environmental science, finance, and engineering frequently involve imbalanced regression tasks, and we need more robust approaches that tackle the problem of imbalanced regression as we have seen in classification.

Adaptive oversampling techniques in classification, such as the method by Wang et al. (2020), which leverages local distribution information to generate synthetic minority instances, demonstrate the potential of these strategies. However, these methods rely on discrete class labels and fixed thresholds that do not translate well to regression tasks, where the target variable is continuous.

Our method, LDAO, addresses these limitations by preserving data structure across all target ranges. Instead of using global thresholds or discarding frequent samples, LDAO augments each cluster of the target distribution independently, generating synthetic data that match local distribution patterns. Our goal in LDAO is to improve the model's prediction of rare samples while enhancing its overall generalization. LDAO aims to mitigate the trade-off between emphasizing rare samples and preserving overall generalization ability.

## 4 LDAO Method

We now describe LDAO, our local distribution-based adaptive oversampling method that effectively addresses the challenges of imbalanced regression. LDAO decomposes the global distribution using $k$-means clustering. This clustering is applied simultaneously to both input features and target values, creating a mixture of local distributions in a joint feature–target space.

After clustering, we apply Gaussian kernel density estimation (KDE) within each cluster to model the local distribution independently and generate synthetic data points that closely match the cluster's actual distribution. By drawing samples independently within each cluster, LDAO avoids creating unrealistic synthetic points. Finally, we merge these augmented clusters into a single augmented dataset. The entire

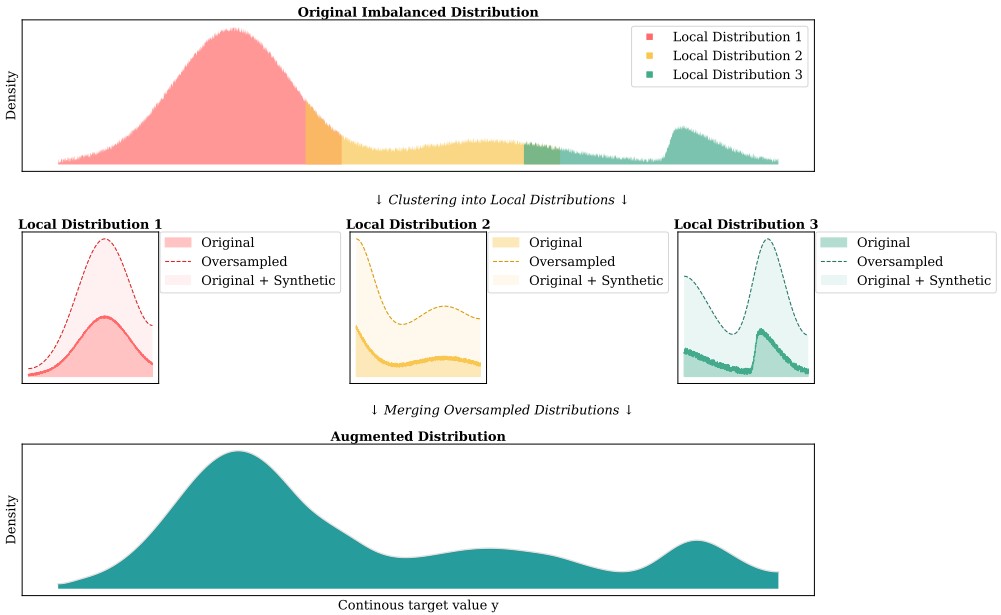

Figure 3: LDAO process overview. First, the imbalanced dataset (top) is decomposed into clusters. Then, each cluster is oversampled individually using kernel density estimation (middle). Lastly, these balanced clusters are combined into one dataset (bottom).

workflow is summarized in Figure 3: data clustering, local density estimation for synthetic data generation, and merging to achieve an augmented overall distribution.

Given a dataset $\mathcal{D} = \{(\mathbf{x}_i, y_i)\}_{i=1}^{N}$, where $\mathbf{x}_i \in \mathbb{R}^d$ represents features and $y_i \in \mathbb{R}$ represents the target variable, imbalance typically manifests as regions with fewer observations, making these areas difficult to model accurately. LDAO models the joint distribution of features and targets. Each data point $(\mathbf{x}_i, y_i)$ is embedded in the combined feature-target space as $\mathbf{z}_i = (\mathbf{x}_i, y_i) \in \mathbb{R}^{d+1}$. The overall joint distribution $P_{X,Y}$ is approximated by a mixture model composed of localized distributions:

$$P_{X,Y}(\mathbf{x}, y) \approx \sum_{k=1}^{K} \pi_k \, P_k(\mathbf{x}, y)$$

where $\pi_k$ denotes the proportion of data in cluster $k$, and $P_k$ characterizes the local distribution of that cluster.

## 4.1 Clustering in the Joint Feature-Target Space

We apply the standard $k$-means clustering algorithm to the dataset to identify natural clusters in the joint feature–target space. Figure 4 illustrates clustering for a sample data set, projected onto the first three principal components, and shows how points with similar features and target values are grouped for more precise local density modelling. For illustration purposes only, we chose 3 as the number of clusters. In this method, every data point $\mathbf{z}_i$ is assigned to one of $K$ clusters. The goal is to find the best set of cluster centroids $\{\boldsymbol{\mu}_k\}_{k=1}^{K}$ and assignments $\{c_i\}_{i=1}^{N}$ such that the total squared Euclidean distance between the data points and their assigned centroids is minimized. Mathematically, we solve:

$$\underset{\{\boldsymbol{\mu}_k\}_{k=1}^{K}, \{c_i\}_{i=1}^{N}}{\arg\min} \sum_{i=1}^{N} \|\mathbf{z}_i - \boldsymbol{\mu}_{c_i}\|^2.$$

This formulation ensures that points with similar characteristics are grouped together (Sugar & James, 2003; Ikotun et al., 2023).

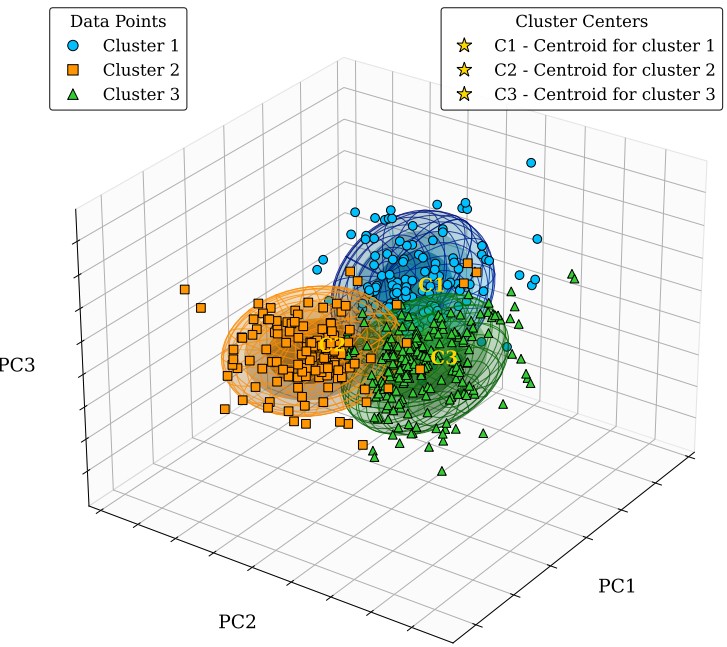

Figure 4: K-means clustering in the joint feature–target space for the Boston dataset (Harrison & Rubinfeld, 1978), projected onto the first three principal components. For illustration only, points are projected onto the first three principal components; clustering itself is done in the full $(d + 1)$-dimensional space. Data points are grouped into three clusters (each indicated by a unique marker and color). Ellipsoidal density contours characterize the spread and orientation of each cluster. The centroids, marked with prominent gold symbols, represent the mean positions of the data points within their respective clusters.

## 4.2 Determining the Number of Clusters

The performance of $k$-means clustering largely depends on the chosen number of clusters $K$. To choose an optimal $K$, we evaluate the Sum of Squared Errors (SSE), defined as:

$$\text{SSE}(K) = \sum_{k=1}^{K} \sum_{\mathbf{z} \in \mathcal{D}_k} \|\mathbf{z} - \boldsymbol{\mu}_k\|^2,$$

where $\mathcal{D}_k$ represents the set of data points in cluster $k$. As we increase $K$, the SSE typically decreases because each point is closer to its cluster center.

To mitigate the pitfalls of the trade-off between oversimplification and excessive clustering, we use the elbow method (Syakur et al., 2018; Nainggolan et al., 2019). This method involves calculating the relative reduction in SSE when increasing the number of clusters. We define the relative improvement $\Delta(K)$ as:

$$\Delta(K) = \frac{\text{SSE}(K-1) - \text{SSE}(K)}{\text{SSE}(K-1)}, \quad K = 2, \ldots, K_{\max}.$$

The optimal number of clusters $(K^*)$ is typically found where there is a significant drop in $\Delta(K)$. One important note is that at the elbow point, adding more clusters does not considerably reduce the Sum of Squared Errors (SSE) and might cause overfitting by capturing noise rather than true underlying patterns (Syakur et al., 2018; Nainggolan et al., 2019).

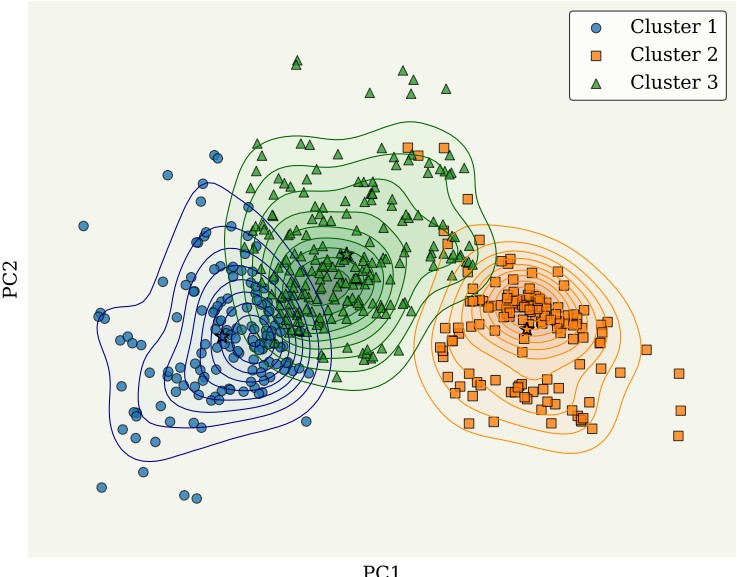

Figure 5: Clusters obtained via $k$-means on the feature–target data are projected onto two principal components. For each cluster, kernel density estimation (KDE) is performed and the overlaid contour lines represent the density gradients of the data in the reduced space.

### 4.3 Kernel Density Estimation in Each Cluster

After the clustering step, we model the local distributions within each cluster separately using kernel density estimation (KDE) (Silverman, 1986), as illustrated in Figure 5.

This approach preserves each cluster's local structure, maintaining accurate relationships between features and targets without mixing patterns across clusters. Let each cluster $\mathcal{D}_k$ contain $n_k$ data points in the joint feature-target space:

$$\mathcal{D}_k = \{\mathbf{z}_1^{(k)}, \mathbf{z}_2^{(k)}, \ldots, \mathbf{z}_{n_k}^{(k)}\},$$

where each data point is defined as $\mathbf{z}_j^{(k)} = (\mathbf{x}_j^{(k)}, y_j^{(k)}) \in \mathbb{R}^{d+1}$. We estimate the local density of cluster $k$ using KDE with a Gaussian kernel as follows:

$$\hat{f}_k(\mathbf{z}) = \frac{1}{n_k} \sum_{j=1}^{n_k} K(\mathbf{z} - \mathbf{z}_j^{(k)}), \tag{1}$$

where the Gaussian kernel $K(\cdot)$ is defined by:

$$K(\mathbf{u}) = \frac{1}{(2\pi)^{\frac{d+1}{2}} |\mathbf{H}|^{1/2}} \exp\left(-\frac{1}{2}\mathbf{u}^\top \mathbf{H}^{-1} \mathbf{u}\right), \quad \mathbf{u} \in \mathbb{R}^{d+1}.$$

Here, $\mathbf{H}$ is a $(d+1) \times (d+1)$ bandwidth (covariance) matrix that controls the shape and smoothness of the density estimate (Scott, 2015). The bandwidth matrix $\mathbf{H}$ determines how closely the estimated density follows the observed data points. A smaller determinant $|\mathbf{H}|$ results in sharply peaked density estimates localized around data points, while a larger determinant yields smoother densities.

Selecting an appropriate bandwidth matrix $\mathbf{H}$ is critical to KDE performance (Sheather & Jones, 1991). If $|\mathbf{H}|$ is too small, KDE tends to overfit noise, resulting in unrealistic density estimates. Conversely, an excessively large $|\mathbf{H}|$ oversmooths important local features. Practical approaches such as cross-validation, plug-in estimators, or generalized Silverman's rule for multivariate KDE can be employed independently for each cluster (Zhang et al., 2006). By fitting KDE independently to each cluster using carefully selected bandwidth matrices, we create localized models that accurately represent each region's unique distribution.

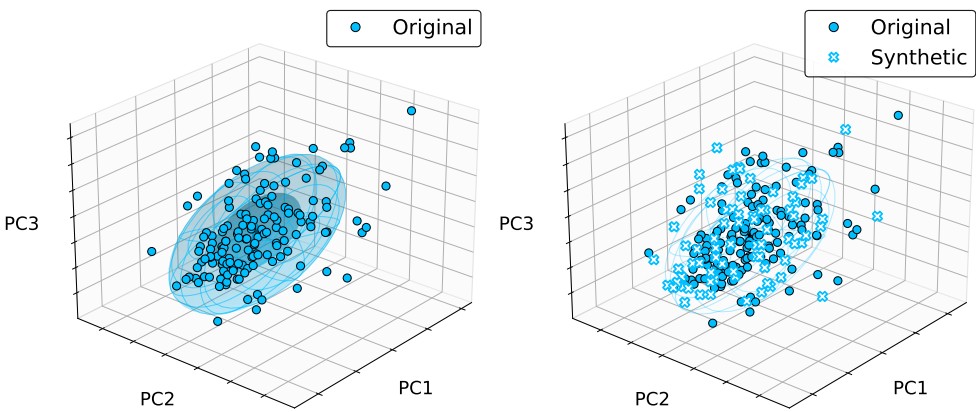

Figure 6: Each cluster is independently oversampled based on its own density estimate, ensuring that both sparse and dense areas are adequately represented without altering their local characteristics; the same procedure is applied to the remaining clusters, with cluster 1 displayed here.

This ensures that synthetic data generated in the subsequent steps reflect genuine local patterns, effectively addressing imbalance without distorting critical data relationships.

## 4.4 Oversampling Clusters

Once the density of each cluster has been estimated, we independently oversample each cluster. Figure 6 illustrates this process. Each cluster $k$, originally containing $n_k$ points, is expanded by generating synthetic data points until reaching a target size defined as:

$$n'_k = \lceil \alpha_k n_k \rceil,$$

where the multiplier $\alpha_k > 1$ is a parameter of the algorithm that determines how much each cluster grows. We generate exactly $n'_k - n_k$ synthetic points $\mathbf{z}^*$ drawn from the local KDE-based density estimate $\hat{f}_k(\mathbf{z})$. The multiplier $\alpha_k$ can be uniform (identical across clusters) or adaptive (higher for clusters that are smaller or less dense). Typically, smaller or sparser clusters receive a larger multiplier, ensuring balanced representation throughout the dataset.

Synthetic points $\mathbf{z}^*$ are generated directly from the KDE-based local distribution defined in Equation equation 1:

$$\mathbf{z}^* \sim \hat{f}_k(\mathbf{z}).$$

For KDE with a Gaussian kernel using bandwidth matrix $\mathbf{H}$, synthetic samples are generated by randomly selecting an existing cluster data point and adding a Gaussian perturbation scaled by the Cholesky decomposition of the bandwidth matrix:

$$\mathbf{z}^* = \mathbf{z}_j^{(k)} + \mathbf{H}^{1/2}\boldsymbol{\epsilon}, \quad \text{where} \quad j \sim \text{Uniform}\{1, \ldots, n_k\}, \quad \boldsymbol{\epsilon} \sim \mathcal{N}(\mathbf{0}, \mathbf{I}_{d+1}).$$

Here, $\mathbf{H}^{1/2}$ is the matrix square root (typically via Cholesky decomposition) of the bandwidth covariance matrix $\mathbf{H}$. This procedure accurately and efficiently samples synthetic points from the KDE estimate, reflecting the local structure and covariance of each cluster(Silverman, 1986; Scott, 2015).

## 4.5 Merging Augmented Clusters

After independently oversampling each cluster, the resulting augmented clusters are merged into a single balanced dataset, as illustrated in Figure 7. Formally, each augmented cluster is defined as:

$$\hat{\mathcal{D}}_k = \mathcal{D}_k \cup \mathcal{D}_k^*,$$

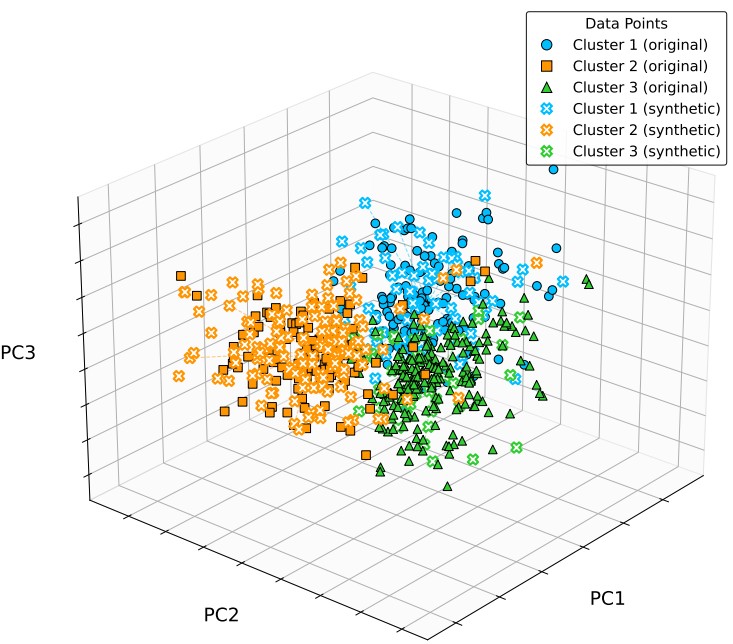

Figure 7: Visualization of original (filled markers) and synthetic (outlined markers) points in each cluster after local oversampling. The merged dataset is augmented while preserving all original data.

where $\mathcal{D}_k^*$ represents the synthetic samples generated for cluster $k$. The final augmented dataset $\hat{\mathcal{D}}$ is constructed by merging all augmented clusters:

$$\hat{\mathcal{D}} = \bigcup_{k=1}^{K} \hat{\mathcal{D}}_k.$$

The size of each cluster after augmentation is explicitly given by:

$$|\hat{\mathcal{D}}_k| = n_k' = \lceil \alpha_k n_k \rceil.$$

Hence, the total size of the merged dataset becomes:

$$|\hat{\mathcal{D}}| = \sum_{k=1}^{K} n_k' = \sum_{k=1}^{K} \left( n_k + (n_k' - n_k) \right).$$

By carefully selecting the multipliers $\alpha_k$, we precisely control the number of synthetic points contributed by each cluster. This yields an augmented representation across the entire range of frequent and rare target values. The complete LDAO procedure is presented in Algorithm 1, which outlines all four steps of our proposed method.

## 5 Evaluation methodology

We evaluated LDAO with current state-of-the-art imbalanced regression approaches using a diverse collection of benchmark datasets. This section details our experimental design, including dataset selection, baseline methods, implementation specifics, hyperparameter tuning, and validation procedures.

---

**Algorithm 1** Local distribution–based adaptive oversampling (LDAO)

---

1: **Input:** data $\{(x_i, y_i)\}_{i=1}^N$, $K_{\min}, K_{\max}$, factors $\{\alpha_k\}$
2: **Output:** augmented set $\widehat{\mathcal{D}}$
3: Embed $z_i \leftarrow (x_i, y_i)$
4: **for** $K = K_{\min}$ **to** $K_{\max}$ **do**
5:     Cluster $\{z_i\} \rightarrow \{\mathcal{D}_k\}$; compute $\text{SSE}(K)$
6: **end for**
7: Select $K^*$ via elbow; recluster into $\{\mathcal{D}_k\}_{k=1}^{K^*}$
8: **for** $k = 1$ **to** $K^*$ **do**
9:     Fit KDE $f_k$ on $\mathcal{D}_k$
10:     Sample $\lceil \alpha_k |\mathcal{D}_k| \rceil - |\mathcal{D}_k|$ points from $f_k$ to form $\widehat{\mathcal{D}}_k$
11: **end for**
12: $\widehat{\mathcal{D}} \leftarrow \bigcup\limits_{k=1}^{K^*} \widehat{\mathcal{D}}_k$
13: **return** $\widehat{\mathcal{D}}$

---

Table 1: Dataset characteristics showing number of instances, features, and size category.

| Dataset | Instances | Features | Size | Dataset | Instances | Features | Size |
|---|---|---|---|---|---|---|---|
| A1 | 198 | 11 | Small | DEBUTANIZER | 2394 | 7 | Large |
| A2 | 198 | 11 | Small | DEE | 365 | 6 | Small |
| A3 | 198 | 11 | Small | DIABETES | 43 | 2 | Small |
| A7 | 198 | 11 | Small | ELE-1 | 495 | 2 | Small |
| ABALONE | 4177 | 8 | Large | ELE-2 | 1056 | 4 | Medium |
| ACCELERATION | 1732 | 14 | Medium | FORESTFIRES | 517 | 12 | Medium |
| AIRFOILD | 1503 | 5 | Medium | FRIEDMAN | 1200 | 5 | Medium |
| ANALCAT | 450 | 11 | Small | FUEL | 1764 | 37 | Medium |
| AUTOMPG6 | 392 | 5 | Small | HEAT | 7400 | 11 | Large |
| AUTOMPG8 | 392 | 7 | Small | HOUSE | 22784 | 16 | Large |
| AVAILABLE_POWER | 1802 | 15 | Medium | KDD | 316 | 18 | Small |
| BASEBALL | 337 | 16 | Small | LASER | 993 | 4 | Medium |
| BOSTON | 506 | 13 | Medium | LUNGCANCER | 442 | 24 | Small |
| CALIFORNIA | 20640 | 8 | Large | MACHINECPU | 209 | 6 | Small |
| COCOMO | 60 | 56 | Small | CONCRETE_STRENGTH | 1030 | 8 | Medium |
| COMPACTIV | 8192 | 21 | Large | META | 528 | 65 | Medium |
| MORTGAGE | 1049 | 15 | Medium | MAXIMAL_TORQUE | 1802 | 32 | Medium |
| PLASTIC | 1650 | 2 | Medium | CPU | 8192 | 12 | Large |
| POLE | 14998 | 26 | Large | TRIAZINES | 186 | 60 | Small |
| QUAKE | 2178 | 3 | Large | WANKARA | 1609 | 9 | Medium |
| SENSORY | 576 | 11 | Medium | WINE_QUALITY | 1143 | 12 | Medium |
| STOCK | 950 | 9 | Medium | WIZMIR | 1461 | 9 | Medium |
| TREASURY | 1049 | 15 | Medium | | | | |

## 5.1 Datasets

We evaluated our method using 45 datasets from three sources: the Keel repository (Alcalá-Fdez et al., 2011), the collection at `https://paobranco.github.io/DataSets-IR` (Branco et al., 2019), and the repository at `https://github.com/JusciAvelino/imbalancedRegression` (Avelino et al., 2024).

These datasets span multiple domains and vary in size, dimensionality, and degree of imbalance, making them standard benchmarks for the rigorous evaluation of imbalanced regression methods and enabling fair comparisons with existing approaches. Table 1 shows the number of instances and features for each dataset. For further analysis in the results section, we categorized the datasets based on their number of instances. Specifically, datasets with fewer than 500 instances are labeled as "Small", those with between 500 and 1999 instances as "Medium", and those with more than 1999 instances as "Large". This categorization facilitates a detailed evaluation of our method's performance relative to dataset size.

## 5.2 Metrics

We evaluate LDAO against state-of-the-art approaches using multiple metrics that measure performance on both frequent and rare target values.

### 5.2.1    Root Mean Square Error (RMSE)

Root Mean Square Error (RMSE) measures the overall prediction accuracy by calculating the square root of the average squared difference between predicted values and actual observations:

$$\text{RMSE} = \sqrt{\frac{1}{n} \sum_{i=1}^{n} (y_i - \hat{y}_i)^2}, \tag{2}$$

where $y_i$ represents the true target value and $\hat{y}_i$ the predicted value for the $i$-th instance. RMSE provides a general assessment of model performance but can be dominated by errors in densely populated regions.

### 5.2.2    Squared Error-Relevance Area (SERA)

This metric provides a flexible way to evaluate models under non-uniform domain preferences. Let $\phi(\cdot) \colon \mathcal{Y} \to [0,1]$ be a relevance function that assigns higher scores to more important (for example, rare or extreme) target values. Then for any relevance threshold $t$, let

$$D^t = \{(x_i, y_i) \mid \phi(y_i) \geq t\},$$

and define

$$SER_t = \sum_{(x_i, y_i) \in D^t} \left( \hat{y}_i - y_i \right)^2. \tag{3}$$

SERA then integrates this quantity over all $t \in [0,1]$:

$$SERA = \int_0^1 SER_t \, dt = \int_0^1 \sum_{(x_i, y_i) \in D^t} \left( \hat{y}_i - y_i \right)^2 dt. \tag{4}$$

SERA weights prediction errors by $\phi(y_i)$, emphasizing performance on extreme values while still considering accuracy across the entire domain. This makes it well-suited for imbalanced regression, where predicting rare values accurately is crucial (Ribeiro & Moniz, 2020).

### 5.2.3    Mean Absolute Error (MAE)

Mean Absolute Error (MAE) is an alternative error metric that evaluates the overall prediction performance by averaging the absolute differences between the predicted and actual values:

$$\text{MAE} = \frac{1}{n} \sum_{i=1}^{n} |y_i - \hat{y}_i| . \tag{5}$$

MAE is less sensitive to outliers compared to RMSE and provides a straightforward interpretation of the average prediction error.

### 5.3    Machine Learning Algorithms

Following the approach of (Steininger et al., 2021), we evaluated all methods using a Multi-Layer Perceptron (MLP) with three hidden layers (10 neurons each) and ReLU activations. The output layer uses linear activation for regression. We trained models for 1000 epochs using Adam optimizer with early stopping to prevent overfitting. We compared LDAO against four approaches: Baseline (no resampling, using the original imbalanced data), SMOGN (an extension of SMOTER that incorporates Gaussian noise during oversampling), G-SMOTE (Geometric SMOTE adapted for regression tasks, using geometric interpolation), and DenseLoss (a cost-sensitive approach that weights errors by target density). These methods represent the current state-of-the-art method for handling imbalanced regression.

Table 2: Hyperparameter search spaces for each compared method

| Method | Hyperparameter | Range | Description |
|---|---|---|---|
| LDAO | K (candidate clusters) | [2, 6] | Number of clusters tested (optimal K is selected using the elbow method) |
| | Multiplier per cluster | [1.0, 3.0] | Oversampling factor |
| | Bandwidth per cluster | [0.1, 2.0] | KDE smoothing parameter |
| SMOGN | k | {3, 5, 7, 9} | Number of neighbors |
| | sampling_method | {extreme, balance} | Sampling approach |
| | rel_thres | [0.0, 1.0] | Relevance threshold |
| G-SMOTE | q_rare | [0.05, 0.25] | Quantile for rarity |
| | truncation_factor | [0.0, 0.9] | Geometric truncation |
| | deformation_factor | [0.0, 0.9] | Geometric deformation |
| | k_neighbors | [2, 5] | Number of neighbors |
| | oversampling_factor | [1.0, 5.0] | Amount of oversampling |
| DenseLoss | alpha | [0.0, 2.0] | Density weighting factor |

## 5.4 Implementation Resources

For our evaluation metrics, we utilized the SERA implementation from the ImbalancedLearningRegression Python package (Wu et al., 2022). The SMOGN method was implemented using the package developed by Kunz (Kunz, 2020). We implemented the DenseLoss and G-SMOTE methods based on their original papers, carefully following the authors' descriptions and guidelines to ensure faithful reproduction of their approaches. To optimize the hyperparameters for each method, we employed the Optuna framework (Akiba et al., 2019). Optuna leverages Bayesian optimization using the Tree-structured Parzen Estimator (TPE) sampler, which efficiently balances exploration and exploitation during the search process. By tuning hyperparameters separately for each dataset, our approach accommodates the unique statistical properties and distribution characteristics inherent to each dataset, ensuring optimal performance for all methods.

## 5.5 Experimental Framework

We employed 5 runs of 5-fold cross-validation for all experiments. This outer fold cross-validation divided each dataset into five equal portions, with each fold using four portions (80% of data) for training and one portion (20% of data) as the test set. Each data portion served as a test set exactly once across the five folds in each run. For hyperparameter tuning within each fold, we further divided the training data into sub-training (80%) and validation (20%) sets. We utilized Bayesian optimization with 15 trials to efficiently search the parameter space, as it generally finds better hyperparameter values than a grid search while requiring fewer evaluations.

Table 2 presents the hyperparameters and search ranges for each method. LDAO's parameters include the oversampling multiplier and KDE bandwidth. SMOGN uses neighborhood size, sampling approach, and relevance threshold. G-SMOTE involves quantile for rarity, truncation factor, deformation factor, number of neighbors, and oversampling factor. DenseLoss works with the density weighting parameter.

## 6 Results

We present the results of comparing LDAO with state-of-the-art oversampling methods across various performance dimensions. In Figure 8, we conduct pairwise comparisons for each evaluation metric (RMSE, SERA, MAE) to demonstrate how LDAO performs relative to each competing method individually. RMSE and MAE evaluate overall model performance across the entire target distribution, while SERA places greater emphasis on errors associated with rare samples.

For each dataset, a winner is determined based on the outcomes over 25 folds, meaning that each dataset contributes one win, and there are 45 wins in total. Specifically, within each dataset, the method that prevails in the majority of the 25 folds is declared the winner. This procedure is repeated against each competing method. Next, we perform the Wilcoxon Signed-Rank Test to compare LDAO's performance vector with that of the competing method. In this test, we compute the differences between the 25-fold loss

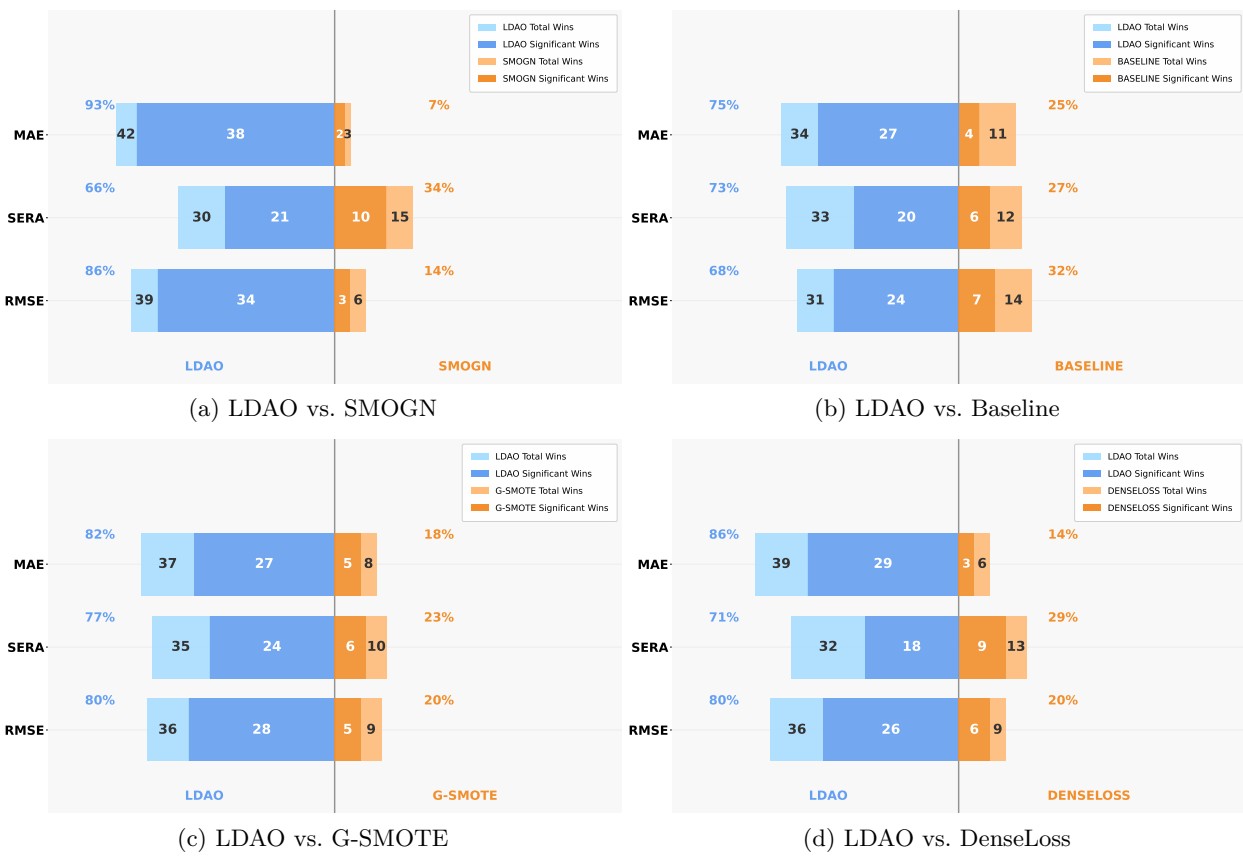

Figure 8: Pairwise comparisons of LDAO with SMOGN, Baseline, G-SMOTE, and DenseLoss across RMSE, SERA, and MAE. Light-colored bars indicate total wins, while darker segments represent statistically significant wins ($\alpha = 0.05$).

metric values of LDAO and those of the competing method to construct a difference vector. The Wilcoxon test then examines whether these differences are statistically significant, using an alpha level of 0.05, which indicates a 5% risk of incorrectly concluding that a difference exists when there is none (Wilcoxon, 1945).

In Figure 8, the lighter color represents the total number of wins for each method, while the darker portion indicates the number of wins that are statistically significant. The figure illustrates that LDAO outperforms all competing methods across all evaluated metrics. Although SMOGN exhibits lower performance in terms of RMSE and MAE, its SERA measurement shows significantly superior performance in more tests, aligning with its emphasis on rare samples; nevertheless, SMOGN remains inferior to LDAO overall. The other methods display a more consistent performance across the different metrics.

Table 3: Mean rank and counts of best (rank 1) and worst (rank 5) across 45 datasets. The lowest mean ranks are highlighted in bold black, the highest rank 1 counts in bold green, and the highest rank 5 counts in bold red.

| Method | RMSE | | | SERA | | | MAE | | |
|---|---|---|---|---|---|---|---|---|---|
| | Mean | R1 | R5 | Mean | R1 | R5 | Mean | R1 | R5 |
| **LDAO** | **2.151** | **27** | 4 | **2.491** | **25** | 7 | **2.001** | **29** | 2 |
| BASELINE | 2.817 | 9 | 1 | 3.106 | 2 | 5 | 2.843 | 6 | 1 |
| DENSELOSS | 3.257 | 3 | 12 | 3.118 | 4 | 10 | 3.353 | 0 | 11 |
| G-SMOTE | 3.025 | 5 | 5 | 3.286 | 1 | **14** | 2.867 | 7 | 2 |
| SMOGN | 3.749 | 1 | **22** | 2.998 | 10 | 9 | 3.936 | 1 | **28** |
| **Total** | – | 45 | 44 | – | 42 | 45 | – | 43 | 44 |

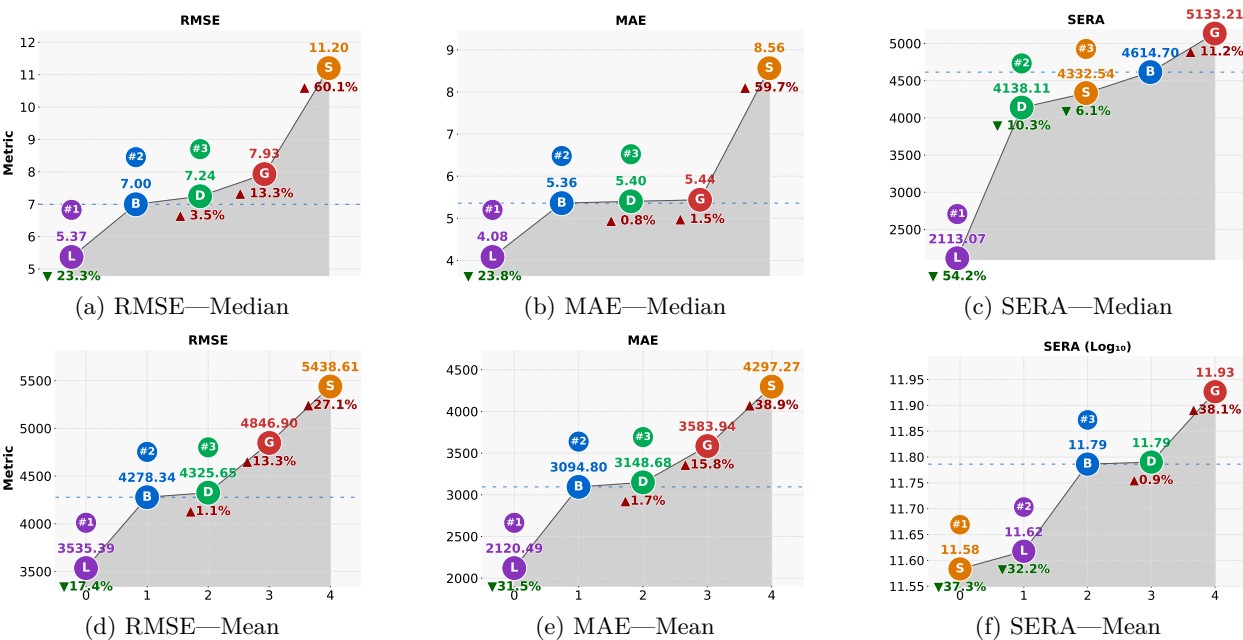

Figure 9: Comparison of methods across median and mean values of RMSE, MAE, and SERA.

In Figure 9, we computed both the median and the mean of various metrics across all datasets for each method. In this analysis, the methods were evaluated collectively rather than in pairwise comparisons. The blue circle represents the baseline benchmark, and the percentages, whether negative or positive, indicate how each oversampling technique performs relative to this baseline.

LDAO again demonstrates robust performance and surpasses its peer methods even when all techniques are assessed together. In each figure, the top three methods are highlighted, including the baseline. The only circumstance in which a peer method, SMOGN, outperformed LDAO occurred when calculating the mean SERA value; in that specific case, SMOGN achieved slightly better results, although both methods significantly outperformed the others. For the SERA mean computation across datasets, a base-10 logarithmic scale was employed to accommodate the high loss values observed on rare samples in some datasets, thereby enhancing clarity.

Table 3 presents the mean rank of each method for RMSE, SERA, and MAE across all 45 datasets. A Rank is the position given to a method when methods are sorted by performance for a given dataset and metric. Rank 1 for the lowest error (best) and Rank 5 for the highest error (worst). For each dataset and metric, a method's rank was first averaged over 25 cross-validation folds; these per-dataset ranks were then averaged across all datasets to produce the Mean column. The R1 column indicates the number of datasets in which a method achieved the best (lowest) rank, and the R5 column indicates the number of datasets in which it achieved the worst (highest) rank. In cases where two or more methods shared virtually identical best or worst values, no count was assigned, so some column totals may sum to less than 45.

As shown in Table 3, LDAO outperforms the other methods, achieving the lowest mean ranks and the highest R1 counts across RMSE, SERA, and MAE, while SMOGN exhibits the highest R5 counts for RMSE and MAE, indicating the weakest performance on those metrics. Baseline, DenseLoss, and G-SMOTE show intermediate performance, with G-SMOTE also recording a high R5 count in SERA.

Figure 10 offers an alternative view by displaying the mean ranking of each method as a function of dataset size. The rankings separate the data sets into small, medium and large based on the classifications previously given in table 1. This evaluation aims to determine how oversampling methods perform across different dataset sizes, as some methods might be more effective on smaller datasets while others excel with larger ones.

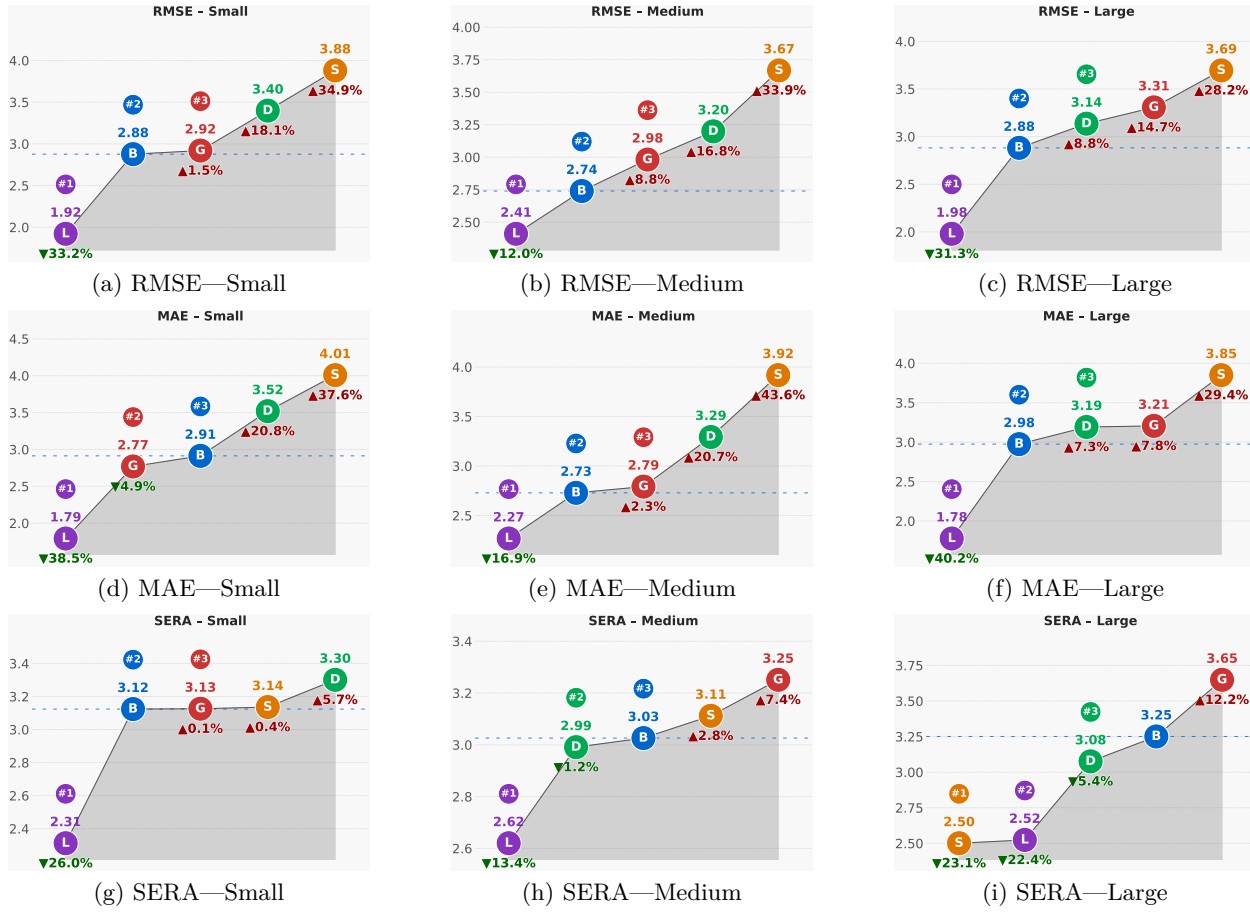

Figure 10: Rank-visualizations for each metric and dataset-size category. Letters denote methods: L=LDAO, B=Baseline, D=DENSELOSS, G=G-SMOTE, S=SMOGN.

LDAO consistently outperformed competing methods, achieving the lowest mean ranking across all metrics and dataset sizes. The only case in which a competing method performed slightly better was for the SERA metric on large datasets, where SMOGN outperformed LDAO; however, LDAO remained superior for both small and medium datasets. In this figure, the blue circle represents the baseline model benchmark, providing a reference for the performance of the various methods.

Table 4: Performance comparison across *QSAR-TID-10624*, *COIL-20*, and *Grocery*. The best values in each column are highlighted in bold dark green.

| Method | QSAR-TID-10624 | | | COIL-20 | | | Grocery | | |
|---|---|---|---|---|---|---|---|---|---|
| | (466 samples, 1024 features) | | | (1440 samples, 1024 features) | | | (22500 samples, 870 features) | | |
| | RMSE | $\log_{10}$(SERA) | MAE | RMSE | $\log_{10}$(SERA) | MAE | RMSE | $\log_{10}$(SERA) | MAE |
| **LDAO** | **0.92** | **0.86** | **0.64** | **1.68** | **0.63** | **0.96** | **655.53** | **8.37** | **495.74** |
| Baseline | 1.00 | 0.90 | 0.71 | 1.83 | 0.70 | 1.09 | 1479.03 | 8.93 | 1292.71 |
| DenseLoss | 1.03 | 0.86 | 0.73 | 1.81 | 0.70 | 1.10 | 1476.75 | 8.93 | 1290.42 |
| G-SMOTE | 0.98 | 0.84 | 0.69 | 1.81 | 0.69 | 1.03 | 1241.74 | 8.82 | 1037.24 |
| SMOGN | 0.98 | 0.84 | 0.69 | 1.83 | 0.69 | 1.08 | 843.49 | 8.41 | 672.88 |

We also evaluated LDAO on high-dimensional tabular datasets (QSAR-TID-10624, COIL-20) and on high-dimensional time-series data (Grocery with an LSTM). Table 4 shows that LDAO again achieves the lowest

Table 5: Stratified performance comparison of LDAO against Baseline, DenseLoss, GSMOTE, and SMOGN across five target bins (B1 sparsest to B5 densest). Bold green entries indicate wins for LDAO.

| Comparison | B1 (Sparsest) | | B2 | | B3 | | B4 | | B5 (Densest) | |
|---|---|---|---|---|---|---|---|---|---|---|
| | LDAO | Other | LDAO | Other | LDAO | Other | LDAO | Other | LDAO | Other |
| vs BASELINE | **28** | 17 | **26** | 19 | **32** | 13 | **35** | 10 | **35** | 10 |
| vs DENSELOSS | **26** | 19 | **27** | 18 | **31** | 14 | **36** | 9 | **39** | 6 |
| vs GSMOTE | **24** | 21 | **31** | 14 | **37** | 8 | **36** | 9 | **34** | 11 |
| vs SMOGN | **24** | 21 | **26** | 19 | **33** | 12 | **38** | 7 | **41** | 4 |

RMSE, SERA, and MAE. In addition, following Steininger et al. (2021), we divide the target range into five equal-width bins, assign each sample to its bin, count and rank them by sample count (Bin 1 is sparsest, Bin 5 is densest), then evaluate performance within each bin to capture method behavior from rare to frequent target regions. As shown in Table 5, LDAO achieves the most wins in bins B3–B5 and maintains leading accuracy in the sparsest bins B1–B2, despite SMOGN and G-SMOTE outperforming it in 21 of 45 comparisons within B1–B2. This confirms that our method not only enhances predictions on rare samples but also significantly improves predictive accuracy in densely sampled regions, yielding robust performance across the entire target distribution.

## 7 Discussion

An interesting observation is the strong performance of the baseline model relative to the oversampling methods. This result suggests that oversampling techniques may underperform without meticulous fine-tuning informed by domain knowledge and underscores the inherent complexity of the imbalanced regression problem compared to classification. In many continuous oversampling approaches, samples are categorized as either rare or frequent; although such a biased assumption can sometimes assist the learning process, it may also lead to diminished performance compared to the baseline. A key attribute of LDAO is its completely data-driven nature. Despite having a few parameters to adjust, LDAO does not require extensive domain expertise, making it readily applicable to a wide range of imbalanced datasets.

Oversampling approaches have the benefit of increasing representation in sparse regions. By generating synthetic samples, these methods can help models learn better from rare target values and improve performance where data is limited. The process involves creating additional data points in underrepresented areas, allowing machine learning models to develop a more complete understanding of the entire target value range. For regression tasks with imbalanced distributions, these techniques provide a practical solution to improve prediction accuracy across the entire spectrum of outcomes.

Specifically for LDAO, the approach leverages local clustering and density estimation to generate synthetic samples that more closely match the underlying data structure. LDAO aims to enhance model performance in regions where simpler approaches might struggle due to data scarcity. The method's focus on local density awareness makes it particularly useful for complex datasets with varying degrees of sparsity. LDAO especially examine sparsity in the joint distribution of features ($X$) and target values ($y$), providing a more detailed and thorough understanding of where data is truly scarce in the multidimensional space.

Despite these advantages, oversampling methods encounter several limitations. They risk overfitting when the synthetic points are too similar to existing samples and may introduce noise if they fail to accurately capture the underlying distribution. Moreover, these techniques typically require additional parameter tuning, such as selecting appropriate rarity thresholds and determining suitable oversampling factors. In LDAO's case, additional considerations include choosing the clustering parameters, specifying the number of synthetic samples, and configuring the density estimation settings. The reliance on k-means partitioning, which assumes roughly spherical clusters, can misassign regions with complex or irregular shapes, while per-cluster KDE bandwidth selection introduces parameter sensitivity and extra computational overhead compared to simpler oversampling schemes. Without proper calibration, synthetic samples might overfit local patterns

or miss critical variations, thereby diminishing the overall effectiveness of the approach when the underlying data characteristics or sparsity patterns are not well captured by the chosen method.

## 8   Conclusion

In this work, we proposed LDAO, a local distribution-based adaptive oversampling method specifically designed to address the challenges of imbalanced regression. By modeling data in a joint feature–target space and generating synthetic samples independently within identified clusters, LDAO effectively preserves the original dataset's statistical structure. Through comprehensive evaluation on a wide variety of datasets, we show that LDAO achieves strong and consistent predictive accuracy, frequently surpassing leading data-level and algorithm-level approaches, particularly where data are sparse.

Oversampling methods, in general, have proven effective at handling imbalance by improving model representation in sparse areas, yet each method has its unique strengths and applications. LDAO contributes to this field by eliminating the need for predefined rarity thresholds and undersampling, thereby preserving valuable information and offering adaptive, data-driven augmentation. As shown in our experiments, these attributes enable LDAO to maintain overall predictive accuracy while enhancing performance in challenging, underrepresented regions. Future work should continue to refine adaptive density-based sampling methods, particularly for datasets with complex, multimodal distributions.

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

## Appendix A: Comparison of Statistical Properties of Synthetic Data

Table 6: Distributional statistics, including mean, standard deviation, skewness, and kurtosis, for each feature and the target variable, comparing the original sparse region (samples for the $\geq$ 75th-percentile tail) to LDAO and SMOGN synthetic samples in the Boston dataset. Values closest to the original distribution are highlighted in bold green.

| Feature | Mean | | | Std Dev | | | Skewness | | | Kurtosis | | |
|---|---|---|---|---|---|---|---|---|---|---|---|---|
| | Orig | LDAO | SMOGN | Orig | LDAO | SMOGN | Orig | LDAO | SMOGN | Orig | LDAO | SMOGN |
| CRIM | -0.314 | **-0.258** | 0.024 | 0.268 | **0.430** | 0.104 | 4.144 | 2.707 | **4.403** | 18.445 | 6.118 | **18.642** |
| ZN | 0.568 | **0.452** | 0.941 | 1.329 | 1.307 | **1.328** | 1.087 | 0.992 | **1.045** | -0.111 | **-0.495** | -0.687 |
| INDUS | -0.678 | **-0.720** | 0.149 | 0.803 | **0.738** | 0.378 | 1.415 | **1.337** | 2.190 | 0.672 | **0.940** | 2.899 |
| CHAS | 0.205 | **0.102** | 0.449 | 1.291 | **1.238** | 1.151 | 2.321 | 2.576 | **2.341** | 3.388 | 4.954 | **3.682** |
| NOX | -0.061 | **-0.246** | 0.190 | 0.688 | **0.911** | 0.286 | -1.768 | **-1.244** | 1.192 | 4.101 | **1.197** | -0.281 |
| RM | 0.210 | **-0.298** | 0.756 | 1.253 | **1.383** | 0.412 | -1.839 | **-1.238** | -0.519 | 1.849 | **-0.223** | -0.505 |
| AGE | -0.474 | **-0.555** | 0.237 | 1.026 | **1.144** | 0.385 | 0.038 | **-0.092** | 1.290 | -1.342 | **-1.330** | 0.012 |
| DIS | 0.219 | **0.093** | 0.576 | 1.073 | **1.259** | 0.710 | 0.517 | **0.583** | 0.972 | 0.213 | **0.278** | -0.240 |
| RAD | -0.385 | **-0.322** | 0.126 | 0.655 | **0.791** | 0.421 | 2.466 | **1.803** | 3.155 | 5.133 | **2.089** | 8.205 |
| TAX | -0.548 | **-0.502** | 0.120 | 0.719 | **0.854** | 0.393 | 1.897 | **1.925** | 3.105 | 3.084 | **2.629** | 7.914 |
| PTRATIO | -0.707 | **-0.485** | 0.066 | 1.004 | **0.926** | 0.197 | -0.333 | **-0.304** | 3.150 | -0.762 | **-0.220** | 8.632 |
| B | 0.164 | **0.122** | 0.540 | 0.912 | **0.964** | 0.228 | -1.459 | **-1.177** | -1.791 | 0.182 | **-0.281** | 1.420 |
| LSTAT | -0.912 | -0.597 | **-1.123** | 0.425 | **0.553** | 0.240 | 1.732 | **1.251** | 0.864 | 4.540 | **1.125** | -0.154 |
| target | 34.638 | **28.409** | 45.990 | 8.017 | 1.620 | **13.949** | 0.808 | -0.140 | **1.304** | -0.611 | **-0.986** | 0.298 |

Table 6 compares the first four moments of each feature and the target between the original data and the synthetic samples generated by LDAO and SMOGN. Values closest to the original distribution are highlighted in green.

LDAO matches the original means for 13 out of 14 entries, while SMOGN's means deviate in 13 out of 14 entries. The standard deviations produced by LDAO have an average absolute deviation of 22.1% from the original values (2 out of 14 within $\pm 5\%$), indicating moderate variance preservation, whereas SMOGN underestimates variability in several features. In terms of skewness, LDAO retains the asymmetry in 10 out of 14 distributions, whereas SMOGN's samples produce the closest skewness in only 4 cases. For kurtosis, LDAO reproduces the tail behavior in 12 out of 14 cases, while SMOGN frequently yields inflated or deflated values because its Gaussian-noise injection fails to capture higher-order moments.

These results directly address the gap of incorrect statistical properties: by preserving means, variances, skewness and kurtosis, LDAO avoids the distortions introduced by linear interpolation. Maintaining accurate marginal statistics is a prerequisite for faithful joint feature–target distributions and for conserving inter-feature correlations, setting the stage for improved overall data fidelity.

## Appendix B: Feature-wise Distribution Comparison: Original vs. LDAO vs. SMOGN for Sparse Region

Figure 11 compares the distribution of each feature and target between the original minority data (cyan, samples above the 75th-percentile), LDAO synthetic data (green) and SMOGN synthetic data (red). Each subplot shows both histogram bars and smooth density curves to highlight how closely each method matches the real data. LDAO closely follows the shape and spread of the original minority samples in nearly every feature. It captures multimodal patterns, skewed tails and the full range of values seen in the real data. In contrast, SMOGN tends to produce narrow, overly regular distributions. Its samples often cluster around central regions and fail to reproduce the true variability of features.

Overall, this analysis shows that LDAO preserves the true characteristics of rare, high-value samples without flattening or narrowing their distributions. SMOGN's linear-interpolation approach, by comparison, smooths away important irregularities and underestimates the natural spread of the minority data.

## Appendix C: Kernel Density Analysis of Target Distribution Preservation

Figure 12 presents density for the target variable, comparing the original minority samples (cyan), LDAO synthetic data (green) and SMOGN synthetic data (red). LDAO closely follows the original density curve across the full target range. It captures the bimodal shape and maintains the correct spread of values, producing synthetic samples that vary naturally and cover all regions seen in the real data. SMOGN, by contrast, concentrates samples in a few intervals and misses other regions entirely. Its synthetic density fails to match the real peaks and troughs of the minority distribution. These results show that LDAO more faithfully preserves the true target distribution, which is crucial for generating reliable synthetic data in imbalanced regression tasks.

## Appendix D: Additional Related Work

Early approaches to imbalanced data focused on data-level methods, particularly in classification, and data-level strategies appear in nearly one-third of imbalanced classification papers, though most were designed for discrete classes (Guo et al., 2017). These methods rebalance datasets by either adding minority samples or removing majority samples (Chawla et al., 2002; He et al., 2008). SMOTE exemplifies this approach by generating synthetic minority samples through interpolation in feature space (Chawla et al., 2002). It works by selecting a minority sample, finding its k-nearest minority neighbors, and creating new points by randomly interpolating between their feature values. This technique helps distribute minority samples more evenly throughout the data space.

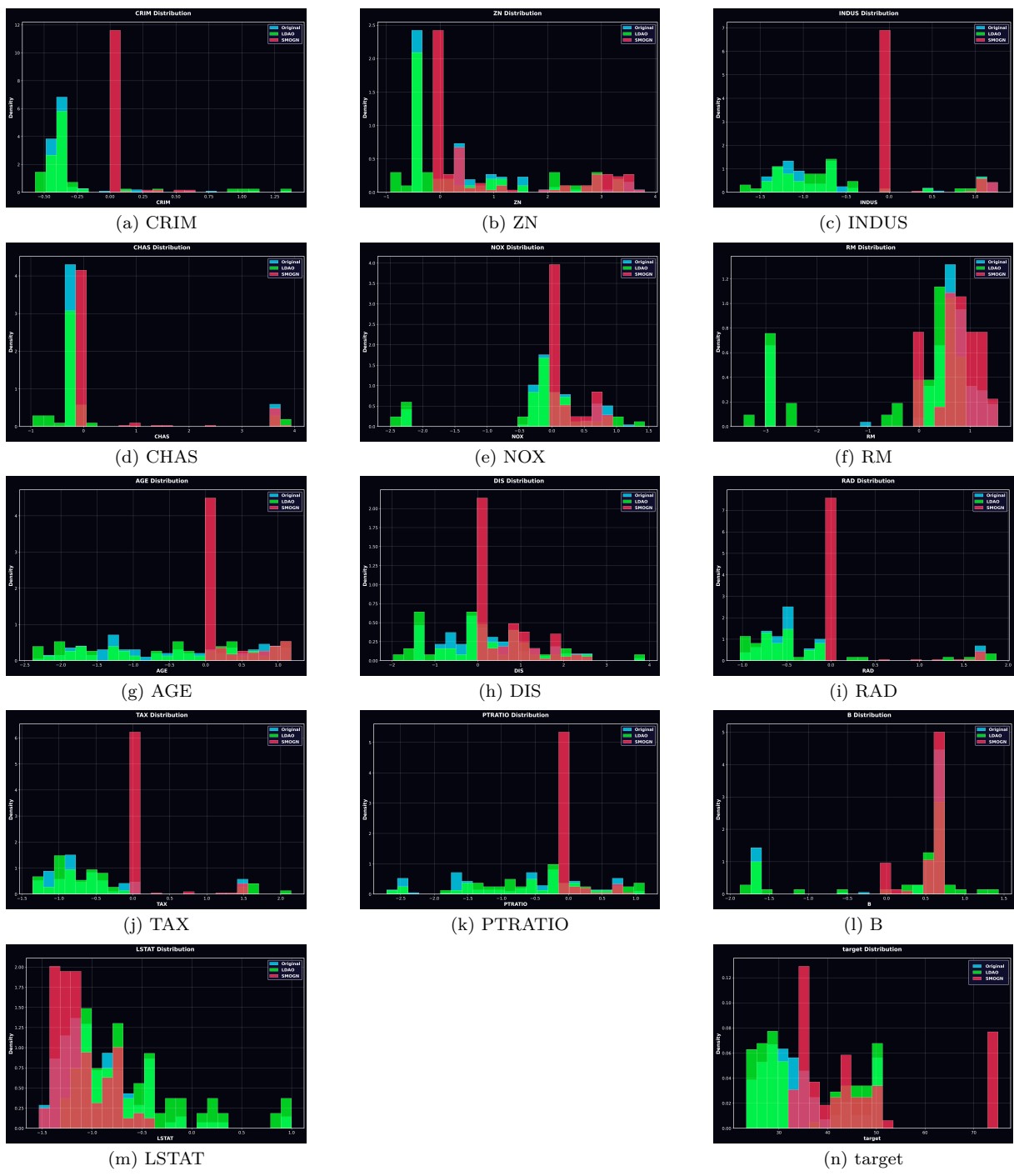

(a) CRIM    (b) ZN    (c) INDUS

(d) CHAS    (e) NOX    (f) RM

(g) AGE    (h) DIS    (i) RAD

(j) TAX    (k) PTRATIO    (l) B

(m) LSTAT    (n) target

Figure 11: Each subplot shows the distribution comparison between the original sparse region (cyan, samples for the ≥ 75th-percentile tail), LDAO synthetic data (green), and SMOGN synthetic data (red) for the respective feature in the Boston dataset.

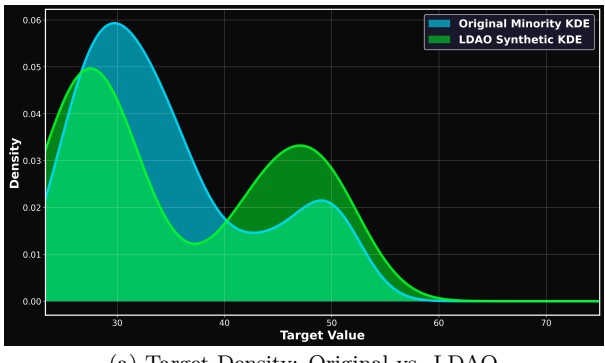 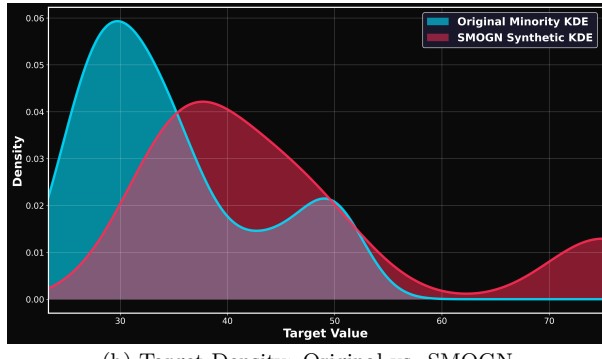

(a) Target Density: Original vs. LDAO

(b) Target Density: Original vs. SMOGN

Figure 12: Kernel density estimation analysis comparing target distribution preservation between original sparse region (cyan, samples for the $\geq$ 75th-percentile tail) and synthetic data generated by LDAO (green) and SMOGN (red) in the Boston dataset.

Several SMOTE variations have emerged to address specific challenges. Borderline-SMOTE focuses on minority samples near class boundaries (Han et al., 2005), while DBSMOTE leverages local density information to generate synthetic points in regions where minority classes are concentrated (Bunkhumpornpat et al., 2012). These methods aim to improve minority class coverage and enhance classifier performance on imbalanced datasets.

Additional regression oversampling approaches include simple random oversampling (replicating high-$\phi$ examples) and pure noise-based oversampling (adding small random perturbations to rare examples) (Branco et al., 2019). Another method, WERCS (Weighted Relevance-based Combination Strategy), probabilistically selects instances for oversampling or undersampling based on relevance scores: rare instances have higher chances of duplication while common instances are typically undersampled unless kept for diversity. This unified approach blends techniques to avoid hard threshold cutoffs (Branco et al., 2019).

Ensemble learning has been explored for imbalanced regression challenges. While boosting and bagging improve model generalization by combining multiple learners, they don't directly address skewed distributions (Hoens & Chawla, 2013). Therefore, researchers typically combine ensemble methods with specific imbalance-countering techniques. One effective approach applies resampling within each ensemble iteration. Resampled Bagging, for instance, uses balanced subsets or weighted samples in each bootstrap replicate, preserving diversity across learners while reducing bias toward majority targets (Branco et al., 2019).

Moniz et al. (2018) extended boosting for rare targets through SMOTEBoost for regression, which generates synthetic samples in extreme target ranges and then uses boosting to emphasize these difficult-to-predict points . This combination of oversampling and ensemble methods proves especially valuable when models need to accurately capture tail values or outliers that standard regression approaches frequently miss.

Evaluation metrics for imbalanced regression have evolved alongside data-level, algorithmic-level, and ensemble-based solutions. Traditional metrics like MSE or MAE are often dominated by errors on frequent cases, prompting the development of specialized metrics that better assess performance on rare targets. The SERA metric (Squared Error Relevance Area) exemplifies this approach by incorporating relevance into evaluation, weighting errors according to the rareness of target values (Ribeiro & Moniz, 2020). Ribeiro and Moniz argue that SERA offers more informative model comparisons in imbalanced domains by highlighting performance on extreme values relative to common ones (Ribeiro & Moniz, 2020). While SERA itself is an evaluation metric rather than a training method, its development reflects the field's increasing focus on accurately assessing how well models handle rare events.

