# OpenReview forum: "Local Distribution-Based Adaptive Oversampling for Imbalanced Regression"
_TMLR — Accepted by TMLR_

### Review · Reviewer_EjBV · 2025-06-08

**Summary Of Contributions:**

This work proposes a new data-level method termed LDAO, for regression tasks in imbalanced data. The method is based on estimating the density in the space of feature-response pairs using a mixture of Gaussian distributions based on the K-means clustering, and then resampling from there. The performance of the proposed method is quantitatively compared with other data-level approaches and baseline, using 45 benchmark dataset. The result show that the proposed method outperforms existing methods in a wide range of tasks.

**Audience:**

Yes

**Broader Impact Concerns:**

Data imbalance is an important topic in statistics. However I do not think that this paper requires adding a Broader Impact Statement as it does not involve verification using data that has as significant social impact.

**Claims And Evidence:**

No

**Requested Changes:**

* The related works sections is a bit like a review. I suggest that the only the works closely related to the proposed method be left in the main text, and the detailed reviews be moved to the appendix as "further related works".
* Please add more detailed discussion of the limitations of the proposed method. For example, density estimation is generally difficult in high-dimensional spaces, but does the proposed method work in such a situation? Also, in the experiment with 45 types of data, the highest score was not achieved in all settings (and this is quite natural and not a drawback), but in what settings was the method inferior to other methods?

**Strengths And Weaknesses:**

## Strengths
* The paper is generally well written; the structure is clear, the procedure of the proposed method is clearly explained.
* Experiments show that the proposed method outperforms existing methods in many settings.

## Weakness
* The related works takes up almost two pages, which is 12 percent of the main text. This lengthy part decrease the readability, although this could be useful for studying imbalanced regression problem.
* The limit of the proposed method is ambiguous, although the possibility of the overfitting in density estimation is discussed in Section 6.  For example, it is difficult to understand what limitations of existing methods the proposed method is expected to solve, and under what conditions it fails to improve performance.

---

> ### Author Response · Authors · 2025-07-05
> **Response to Reviewer Comments**
>
> Thank you for your constructive feedback. The revised paper has been updated to address the concerns raised in the “Weaknesses” and “Request Changes” comments, as outlined below.
>
> 1- "The related works sections is a bit like a review."
>
> Thank you for your suggestion. We agree that keeping work closer to our method and moving the less related ones to the appendix will enhance the readability of the paper. We have revised our related works section (See Appendix D).
>
> 2- " Please add more detailed discussion of the limitations of the proposed method."
>
> We appreciate the reviewer’s suggestion and we have added detailed information about the limitations of LDAO in the discussion section of our revised paper: In LDAO’s case, additional considerations include choosing the clustering parameters, specifying the number of synthetic samples, and configuring the density estimation settings. The reliance on k-means partitioning, which assumes roughly spherical clusters, can misassign regions with complex or irregular shapes, while per-cluster KDE bandwidth selection introduces parameter sensitivity and extra computational overhead compared to simpler oversampling schemes. Without proper calibration, synthetic samples might overfit local patterns or miss critical variations, thereby diminishing the overall effectiveness of the approach when the underlying data characteristics or sparsity patterns are not well captured by the chosen method.  This material has been added on page 17 of the manuscript.
>
> 3- "LDAO’s performance on high-dimensional datasets."
>
> We thank you for highlighting this point, and we acknowledge that evaluating LDAO’s performance on high-dimensional data is essential for assessing its applicability and scalability. As requested, we conducted additional experiments on two high-dimensional datasets (QSAR-TID-10624, COIL-20) and one high-dimensional time series dataset (Grocery), which is trained and evaluated with a LSTM architecture.
>
> For QSAR-TID-10624, COIL-20, we have provided the results with and without principal component analysis (90 % of the total variance) to show that our LDAO method works in high-dimensional settings as well.
>
> [WITHOUT PCA](https://drive.google.com/file/d/1OxqvP-qSx55ZI_7gESdyx1k2VLg3HL1E/view)
>
> [WITH PCA](https://drive.google.com/file/d/1WwTvibNNf84jZ29M4EIAj15exgbmS_sF/view?usp=sharing).
>
> [VARIABILITY CAPTURING REPORT](https://drive.google.com/file/d/1Qu2-zkpLuQK31zBM1TpglCOaJyS_GGqJ/view?usp=sharing)
>
>
> The corresponding content has been added to Section 6 in the revised paper (see Table 4).  As can be observed, Table 4 shows that LDAO has superior performance on these high dimensional data sets.  In the interest of comparison, we have conducted a PCA version of the analysis. However, it demonstrates similar results to the “without PCA”  version and does not add to the analysis. It is not included in the paper.
>
> 4- "highest score was not achieved in all settings (and this is quite natural and not a drawback), but in what settings was the method inferior to other methods".
>
> We sincerely appreciate your comment regarding this. As requested, we conducted a stratified analysis using the binning strategy from Steininger et al. (2021):
>
> -   Binning: We split the target range into five equal-width bins (each covering 20 % of the range), counted how many samples fall in each bin, and ranked them by size.
>
>
> 	-   Bin 1: fewest samples
>
>
> 	-   …
>
>
> 	-   Bin 5: most samples
>
>
> -   Supplementary Table: Results are in the supplementary material, accessible via [STRATIFIED_ANALYSIS](https://drive.google.com/file/d/1r8S7l6rJVU7KDI0_HuvBUgJljJHWKiyl/view?usp=sharing).
>
>
> -   Findings:
>
>
> 	-   Oversampling both sparse and dense regions enhances model generalizability and rare-sample performance.
>
>
> 	-   LDAO outperforms other methods in every bin, with especially large gains in B3, B4 and B5.
>
>
> 	-   Methods that only oversample sparse bins saw reduced generalizability in B4 and B5.
>
>
> Our aim with LDAO was to improve predictive accuracy in sparse regions while preserving generalization in dense regions. As shown in the table, LDAO significantly outperformed competing methods in bins B3, B4, and B5, and also yielded better results in bins B1 and B2, the rarest regions. That said, SMOGN and G-SMOTE outperformed LDAO in bins B1 and B2 on 21 of the 45 datasets. This pattern reflects a trade-off in our design, where balancing density-ratio weighting with cluster-wise sampling yields a unified, well-rounded solution that boosts generalization in dense areas while still enhancing accuracy in rare areas.
>
> The corresponding content has been added to Section 6 in the revised paper.
>
> Thank you for your thoughtful comments and suggestions, which have significantly enhanced the rigor, clarity, and impact of our paper.

---

### Review · Reviewer_LLbv · 2025-06-16

**Summary Of Contributions:**

The paper introduces LDAO (Local Distribution-based Adaptive Oversampling), a novel method for imbalanced regression that targets the challenge of predicting rare continuous target values. Unlike traditional approaches that rely on global thresholds, LDAO performs local modeling by clustering the joint feature–target space using k-means. Within each cluster, kernel density estimation is applied to generate synthetic samples that reflect local data distributions. The method is extensively evaluated on 45 datasets, demonstrating superior performance compared to baselines such as SMOGN, G-SMOTE, and DenseLoss across RMSE, MAE, and SERA metrics.

**Audience:**

Yes

**Claims And Evidence:**

No

**Requested Changes:**

Please see weaknesses

**Strengths And Weaknesses:**

Strengths:

1. The paper introduces an intuitive approach that eliminates the need for arbitrary global thresholds.
2. The comprehensive experimental evaluation covers 45 datasets.
3. The method preserves local statistical structure within each cluster by modeling them independently.

Weaknesses:

1.  The literature review overlooks several recent and highly relevant contributions from 2024–2025, [1-6], making it difficult to contextualize LDAO within the modern landscape of imbalanced regression research. No comparison to these recent approaches is made.

2.  The evaluation does not analyze performance separately for rare versus frequent target regions. Reporting aggregate metrics such as RMSE and MAE without stratified breakdowns makes it unclear whether improvements benefit rare regions or are driven by majority cases.

3. The introduction requires substantial reworking as it fails to establish clear research gaps beyond vague statements like "current solutions remain limited." The motivational examples in wind forecasting, healthcare, and other domains lack proper citations and context, weakening the paper's foundation.

4. The paper provides no theoretical foundation, lacking analysis of convergence properties, optimality conditions, or theoretical guarantees about when and why LDAO should work effectively.

5. The method is demonstrated only on tabular datasets. There is no discussion or exploration of how LDAO might extend to structured data types such as images, time series, or text—modalities where imbalanced regression is also relevant.

6. The method assumes k-means clustering is appropriate for the joint feature-target space without theoretical or empirical justification. This assumption may fail in high-dimensional settings, yet no sensitivity analysis or ablation study assesses the impact of this design choice.

7. Figure 1's contrast between classification and regression imbalance oversimplifies both domains and ignores the substantial overlap. Long-tailed classification problems share similar distributional challenges with regression imbalance

[1] Dong et al. (2025): "Improve Representation for Imbalanced Regression through Geometric Constraints" (CVPR 2025)

[2]  Xiong & Yao (2024): "Deep Imbalanced Regression via Hierarchical Classification Adjustment" (CVPR 2024)

[3] Nejjar et al. (2024): "IM-Context: In-Context Learning for Imbalanced Regression Tasks" (TMLR)

[4] Luo et al. (2024): "Revive re-weighting in imbalanced learning by density ratio estimation" (NeurIPS)

[5] Stocksieker et al. (2024): "Generalized Oversampling for Learning from Imbalanced datasets" (TMLR)

[6] Pu et al. (2025): "Leveraging Group Classification with Descending Soft Labeling for Deep Imbalanced Regression" (AAAI)

---

> ### Author Response · Authors · 2025-07-05
> **Response to review comments**
>
> Thank you for your constructive feedback. The revised paper has been updated to address the concerns raised in the “Weaknesses” and “Request Changes” comments, as outlined below.
>
> 1- “The literature review overlooks several recent and highly relevant contributions.”
>
> We truly appreciate your suggestions on the problem of deep imbalanced regression (DIR) and the valuable references you provided, and we have added them to the literature review in the revised paper. Below, we clarify each work’s relevance to our benchmarks:
>
> [DEEP IMBALANCED REGRESSION](https://drive.google.com/file/d/11xn4DBe_mPp7_4fKTM62aSS2-F7pRW0F/view?usp=sharing)
>
> Typical benchmarks for deep imbalanced regression (DIR):
>
> -   CIFAR-100-LT (IF=100): ~ 20,847 images
>
>
> -   AgeDB-DIR: ~16,400 face images
>
>
> -   IMDB-WIKI-DIR: ~ 202,500 face images
>
> Our benchmarks:
>
> -   Alcalá-Fdez et al. (2011)
>
>
> -   Datasets listed in Table 1 of Section 4
>
>
>
> These valuable contributions and their benchmarks advance deep imbalanced regression (DIR), first described by Yang et al. (ICML 2021), though they have mainly been applied to vision-based continuous-label tasks. Our benchmarks complement this work by focusing on smaller data sets from real-world settings where data collection is costly or rare.
>
> Because DIR is an important aspect of imbalanced regression, we have text in the revised paper that describes the references you provided, now on page five of the manuscript. We appreciate these suggestions; they help make the paper more inclusive.
>
>
> 2- "The evaluation does not analyze performance separately for rare versus frequent target regions."
>
> We thank you for raising this concern. While the SERA loss (Sec. 4.2.2) is specifically tailored to showcase performance on rare samples, a stratified breakdown would clarify where LDAO outperformed across the datasets. As requested, we conducted a stratified analysis using the binning strategy from Steininger et al. (2021):
>
> -   Binning: We split the target range into five equal-width bins (each covering 20 % of the range), counted how many samples fall in each bin, and ranked them by size.
>
>
> 	-   Bin 1: fewest samples
>
>
> 	-   …
>
>
> 	-   Bin 5: most samples
>
>
> -   The performance of the methods on each of these bins is calculated.
>
>
> -   Supplementary Table: Results are in the supplementary material, accessible via [STRATIFIED_ANALYSIS](https://drive.google.com/file/d/1r8S7l6rJVU7KDI0_HuvBUgJljJHWKiyl/view?usp=sharing).
>
>
> The stratified analysis shows that LDAO outperforms other methods in all bins.  The corresponding content has been added to Section 6 (see Table 5) in the revised paper.
>
> 3- "The introduction requires substantial reworking as it fails to establish clear research gaps."
>
> We apologize for the lack of clarity. We have revised the introduction in our updated submission. Please find these clarifications regarding this, which are added to Sections 1, 2, 3 and Appendices A, B and C in the revised paper.  In particular, we have focused on linear interpolation issues, which can create issues with dependence between variables.
>
>
> 4- "LDAO’s performance on high-dimensional and time-series datasets."
>
> We thank you for highlighting this point, and we acknowledge that evaluating LDAO’s performance on high-dimensional data is essential for assessing its applicability and scalability. As requested, we conducted additional experiments on two high-dimensional datasets (QSAR-TID-10624, COIL-20) and one high-dimensional time series dataset (Grocery), which is trained and evaluated with a LSTM architecture.
>
> For QSAR-TID-10624, COIL-20, we have provided the results with and without principal component analysis (90 % of the total variance) to show that our LDAO method works in high-dimensional settings as well.
>
> [WITHOUT PCA](https://drive.google.com/file/d/1OxqvP-qSx55ZI_7gESdyx1k2VLg3HL1E/view)
>
> [WITH PCA](https://drive.google.com/file/d/1WwTvibNNf84jZ29M4EIAj15exgbmS_sF/view?usp=sharing)
>
> [VARIABILITY CAPTURING REPORT](https://drive.google.com/file/d/1Qu2-zkpLuQK31zBM1TpglCOaJyS_GGqJ/view?usp=sharing)
>
>
> The corresponding content has been added to Section 6 in the revised paper (see Table 4).  As can be observed, Table 4 shows that LDAO has superior performance on these high-dimensional data sets.  In the interest of comparison, we have conducted a PCA version of the analysis. However, it demonstrates similar results to the “without PCA”  version and does not add to the analysis. It is not included in the paper.
>
> 5- "Figure 1's contrast between classification and regression"
>
> You’re absolutely right that long-tailed classification and regression share distributional challenges. The corresponding Figure 1 has been modified in the revised paper.
>
> Thank you for your thoughtful comments and suggestions, which have significantly enhanced the rigor, clarity, and impact of our paper.

---

### Review · Reviewer_aaGN · 2025-06-23

**Summary Of Contributions:**

This paper addresses the problem of imbalanced regression by proposing a simple yet intuitive oversampling method based on joint feature-output clustering and kernel density ratio estimation. The central idea is to identify dense and sparse regions in the joint input-output space, then apply region-specific oversampling, with more samples generated in sparse areas and fewer in dense ones. The contribution of the paper is
1. Proposes a region-aware oversampling method for imbalanced regression based on joint feature-output clustering.
2. Estimates kernel density ratios to guide adaptive oversampling in dense vs. sparse regions.
3. Demonstrates effectiveness across 45 low-dimensional regression datasets with consistent improvements.

**Audience:**

Yes

**Claims And Evidence:**

No

**Requested Changes:**

1. Discuss whether the method can be applied to high-dimensional cases. If possible, provide related test to show scalability.
2. Please be specific when criticizing existing works. What exactly is being oversimplified? What part of the "nature" is being distorted?
3. Please explain why you are oversampling in dense regions. In class-imbalanced learning, we usually avoid oversampling the majority class — is your setting different, or is this a design compromise?

**Strengths And Weaknesses:**

Strengths

1. This paper addresses a practically important but relatively underexplored problem: imbalanced regression.

2. The proposed method is simple, intuitive, and shows good performance across 45 regression datasets.


Weaknesses
1. All datasets used in the experiments are of relatively low dimensionality (mostly < 20 features, none exceeding 65). This raises concerns about the method's applicability to real-world, high-dimensional settings.

2. Several claims made against previous work are either imprecise or misleading. For example: The paper criticizes earlier studies for "oversimplifying the complexity of continuous distributions" but fails to state clearly on what specific aspects of the complexity are being ignored. It also states that previous methods "distort the nature of the dataset", while the whole principle of imbalance handling is inherently a kind of distortion to better suit learning objectives. These statement could mislead readers unfamiliar with the literature.

3. While the paper criticizes previous work for being oversimplified, it essentially follows the same high-level idea — dividing the space into dense and sparse regions, and oversampling the sparse ones. This framing is not fundamentally different from what has been done before. What’s more confusing is that the method also oversamples in dense regions. This is counterintuitive — in class imbalance problems, we usually avoid oversampling the majority class. Although the rate is low, it’s not clear why any oversampling should be done there at all to tackle the imbalanced problem. The paper doesn’t provide a convincing explanation.

---

> ### Author Response · Authors · 2025-07-05
> **Response to review comments**
>
> Thank you for your constructive feedback. The revised paper has been updated to address the concerns raised in the “Weaknesses” and “Request Changes” comments, as outlined below.
>
> 1- "LDAO’s performance on high-dimensional."
>
> We thank you for highlighting this point, and we acknowledge that evaluating LDAO’s performance on high-dimensional data is essential for assessing its applicability and scalability. As requested, we conducted additional experiments on two high-dimensional datasets (QSAR-TID-10624, COIL-20) and one high-dimensional time series dataset (Grocery), which is trained and evaluated with a LSTM architecture.
>
> For QSAR-TID-10624, COIL-20, we have provided the results with and without principal component analysis (90 % of the total variance) to show that our LDAO method works in high-dimensional settings as well.
>
> [WITHOUT PCA](https://drive.google.com/file/d/1OxqvP-qSx55ZI_7gESdyx1k2VLg3HL1E/view)
>
> [WITH PCA](https://drive.google.com/file/d/1WwTvibNNf84jZ29M4EIAj15exgbmS_sF/view?usp=sharing)
>
> [VARIABILITY CAPTURING REPORT](https://drive.google.com/file/d/1Qu2-zkpLuQK31zBM1TpglCOaJyS_GGqJ/view?usp=sharing)
>
>
> The corresponding content has been added to Section 6 in the revised paper (see Table 4).  As can be observed, Table 4 shows that LDAO has superior performance on these high-dimensional data sets.  In the interest of comparison, we have conducted a PCA version of the analysis. However, it demonstrates similar results to the “without PCA”  version and does not add to the analysis. It is not included in the paper.
>
> 2- "What exactly is being oversimplified? What part of the 'nature' is being distorted?"
>
> We apologize for any lack of clarity. Please find these clarifications regarding this, which are added to Sections 1, 2, 3 and Appendices A, B and C in the revised paper.  In particular, we have focused on linear interpolation issues, which can create issues with dependence between variables.
>
> Please also see the PCA comparison plot below, which focuses on target values ≥ 75th percentile, and note that in comparison with SMOGN, its synthetic points lie along a narrow interpolation ridge, whereas LDAO samples spread more naturally around that extreme tail [PCA VISUALIZATION](https://drive.google.com/file/d/13ZeDMT3EDZGupfgzo0GHpOsItvkkA9ir/view?usp=sharing) (Boston dataset). This figure is included in the text as Figure 2.
>
> We also conducted multiple statistical property tests to compare the quality of synthetic data generated by LDAO against a linear interpolation–based method. Please see [STATISTICAL PROPERTIES COMPARISON](https://drive.google.com/file/d/1rNMhmnkVJC-7uMIB33RcfJBAtcVLJZt2/view?usp=sharing) (Appendix A), which demonstrates that LDAO’s synthetic samples more closely match the statistical properties of the original minority data than those produced by SMOGN.  This Table is included in Appendix A for added clarity on the properties of the augmented data sets.
>
>
> 3- "Why you are oversampling in dense regions."
>
> Thank you for asking this very important question. This is our design compromise, as previous methods oversample only the sparse region and not the dense part. Our approach is distribution-agnostic, allowing the model to learn the alpha parameters to best suit the amount of oversampling needed in each region.
>
> To make this clearer, we performed a stratified performance analysis using the binning strategy from Steininger et al. (2021), suggested by the reviewers:
>
> -   Binning: We split the target range into five equal-width bins (each covering 20 % of the range), counted how many samples fall in each bin, and ranked them by size.
>
>
> 	-   Bin 1: fewest samples
>
>
> 	-   …
>
>
> 	-   Bin 5: most samples
>
>
> -   Supplementary Table: Results are in the supplementary material, accessible via [STRATIFIED_ANALYSIS](https://drive.google.com/file/d/1r8S7l6rJVU7KDI0_HuvBUgJljJHWKiyl/view?usp=sharing) (Section 6).
>
>
> -   Findings:
>
>
> 	-   Oversampling both sparse and dense regions enhances model generalizability and rare-sample performance.
>
>
> 	-   LDAO outperforms other methods in every bin, with especially large gains in B3, B4 and B5.
>
>
> 	-   Methods that only oversample sparse bins saw reduced generalizability in B4 and B5.
>
>
> As shown in the table, LDAO significantly outperformed competing methods in bins B3, B4, and B5, and also yielded better results in bins B1 and B2, the rarest regions. That said, SMOGN and G-SMOTE outperformed LDAO in bins B1 and B2 on 21 of the 45 datasets. **This pattern reflects a trade-off in our design**, where balancing density-ratio weighting with cluster-wise sampling yields a unified, well-rounded solution that boosts generalization in dense areas while still enhancing accuracy in rare areas.
>
> The corresponding content has been added to Section 6 in the revised paper.
>
> Thank you for your thoughtful comments and suggestions, which have significantly enhanced the rigor, clarity, and impact of our paper.

---

### Decision · Action_Editor_PAr3 · 2025-08-11

**Recommendation:** Accept as is

**Audience:**

Yes

**Audience Explanation:**

There are several pros in this paper. For example, 1) The paper introduces an intuitive approach that eliminates the need for arbitrary global thresholds. The proposed method is simple and easy to understand. 2) The comprehensive experimental evaluation covers 45 datasets. The experiments show that it performs better than existing methods on many test cases. 3) The method preserves local statistical structure within each cluster by modeling them independently.

**Claims And Evidence:**

Yes

**Claims Explanation:**

This paper deals with regression problems when data distribution is imbalanced. There are several pros in this paper. For example, 1) The paper introduces an intuitive approach that eliminates the need for arbitrary global thresholds. The proposed method is simple and easy to understand. 2) The comprehensive experimental evaluation covers 45 datasets. The experiments show that it performs better than existing methods on many test cases. 3) The method preserves local statistical structure within each cluster by modeling them independently.